# GLOBAL INDUCING POINT VARIATIONAL POSTERIORS FOR BAYESIAN NEURAL NETWORKS AND DEEP GAUSSIAN PROCESSES

## ABSTRACT

We derive the optimal approximate posterior over the top-layer weights in a Bayesian neural network for regression, and show that it exhibits strong dependencies on the lower-layer weights. We adapt this result to develop a correlated approximate posterior over the weights at all layers in a Bayesian neural network. We extend this approach to deep Gaussian processes, unifying inference in the two model classes. Our approximate posterior uses learned "global" inducing points, which are defined only at the input layer and propagated through the network to obtain inducing inputs at subsequent layers. By contrast, standard, "local", inducing point methods from the deep Gaussian process literature optimise a separate set of inducing inputs at every layer, and thus do not model correlations across layers. Our method gives state-of-the-art performance for a variational Bayesian method, without data augmentation or tempering, on CIFAR-10 of $86.7\%$.

## 1 INTRODUCTION

Deep models, formed by stacking together many simple layers, give rise to extremely powerful machine learning algorithms, from deep neural networks (DNNs) to deep Gaussian processes (DGPs) (Damianou & Lawrence, 2013). One approach to reason about uncertainty in these models is to use variational inference (VI) (Jordan et al., 1999). VI in Bayesian neural networks (BNNs) requires the user to specify a family of approximate posteriors over the weights, with the classical approach being independent Gaussian distributions over each individual weight (Hinton & Van Camp, 1993; Graves, 2011; Blundell et al., 2015). Later work has considered more complex approximate posteriors, for instance using a Matrix-Normal distribution as the approximate posterior for a full weight-matrix (Louizos & Welling, 2016; Ritter et al., 2018). By contrast, DGPs use an approximate posterior defined over functions — the standard approach is to specify the inputs and outputs at a finite number of "inducing" points (Damianou & Lawrence, 2013; Salimbeni & Deisenroth, 2017).

Critically, these classical BNN and DGP approaches define approximate posteriors over functions that are independent across layers. An approximate posterior that factorises across layers is problematic, because what matters for a deep model is the overall input-output transformation for the full model, not the input-output transformation for individual layers. This raises the question of what family of approximate posteriors should be used to capture correlations across layers. One approach for BNNs would be to introduce a flexible "hypernetwork", used to generate the weights (Krueger et al., 2017; Pawlowski et al., 2017). However, this approach is likely to be suboptimal as it does not sufficiently exploit the rich structure in the underlying neural network. For guidance, we consider the optimal approximate posterior over the top-layer units in a deep network for regression. Remarkably, the optimal approximate posterior for the last-layer weights given the earlier weights can be obtained in closed form without choosing a restrictive family of distributions. In particular, the optimal approximate posterior is given by propagating the training inputs through lower layers to compute the top-layer representation, then using Bayesian linear regression to map from the top-layer representation to the outputs.

Inspired by this result, we use Bayesian linear regression to define a generic family of approximate posteriors for BNNs. In particular, we introduce learned "pseudo-data" at every layer, and compute the posterior over the weights by performing linear regression from the inputs (propagated from

lower layers) onto the pseudo-data. We reduce the burden of working with many training inputs by summarising the posterior using a small number of "inducing" points. We find that these approximate posteriors give excellent performance in the non-tempered, no-data-augmentation regime, with performance on datasets such as CIFAR-10 reaching 86.7%, comparable to SGMCMC (Wenzel et al., 2020). Our approach can be extended to DGPs, and we explore connections to the inducing point GP literature, showing that inference in the two classes of models can be unified.

## 2 METHODS

We consider neural networks with lower-layer weights $\{\mathbf{W}_\ell\}_{\ell=1}^L$, $\mathbf{W}_\ell \in \mathbb{R}^{N_{\ell-1} \times N_\ell}$, and top-layer weights, $\mathbf{W}_{L+1} \in \mathbb{R}^{N_L \times N_{L+1}}$, where the activity, $\mathbf{F}_\ell$, at layer $\ell$ is given by,

$$\mathbf{F}_1 = \mathbf{X}\mathbf{W}_1, \qquad \mathbf{F}_\ell = \phi\left(\mathbf{F}_{\ell-1}\right)\mathbf{W}_\ell \quad \text{for } \ell \in \{2, \ldots, L\}, \qquad (1)$$

where $\phi(\cdot)$ is an elementwise nonlinearity. The outputs, $\mathbf{Y} \in \mathbb{R}^{P \times N_{L+1}}$, depend on the top-level activity, $\mathbf{F}_L$, and the output weights, $\mathbf{W}_{L+1}$, according to a likelihood, $\mathrm{P}\left(\mathbf{Y}|\mathbf{W}_{L+1}, \mathbf{F}_L\right)$. In the following derivations, we will focus on $\ell > 1$; corresponding expressions for the input layer can be obtained by replacing $\phi(\mathbf{F}_0)$ with the inputs, $\mathbf{X} \in \mathbb{R}^{P \times N_0}$. The prior over weights is independent across layers and output units (see Sec. 2.3 for the form of $\mathbf{S}_\ell$),

$$\mathrm{P}\left(\mathbf{W}_\ell\right) = \textstyle\prod_{\lambda=1}^{N_\ell} \mathrm{P}\left(\mathbf{w}_\lambda^\ell\right), \qquad \mathrm{P}\left(\mathbf{w}_\lambda^\ell\right) = \mathcal{N}\left(\mathbf{w}_\lambda^\ell \,\middle|\, \mathbf{0}, \tfrac{1}{N_{\ell-1}}\mathbf{S}_\ell\right), \qquad (2)$$

where $\mathbf{w}_\lambda^\ell$ is a column of $\mathbf{W}_\ell$, i.e. all the input weights to unit $\lambda$ in layer $\ell$. To fit the parameters of the approximate posterior, $\mathrm{Q}\left(\{\mathbf{W}\}_{\ell=1}^{L+1}\right)$, we maximise the evidence lower bound (ELBO),

$$\mathcal{L} = \mathbb{E}_{\mathrm{Q}\left(\{\mathbf{W}\}_{\ell=1}^{L+1}\right)}\left[\log \mathrm{P}\left(\mathbf{Y}, \{\mathbf{W}\}_{\ell=1}^{L+1}|\mathbf{X}\right) - \log \mathrm{Q}\left(\{\mathbf{W}\}_{\ell=1}^{L+1}\right)\right]. \qquad (3)$$

To build intuition about how to parameterise $\mathrm{Q}\left(\{\mathbf{W}\}_{\ell=1}^{L+1}\right)$, we consider the optimal $\mathrm{Q}\left(\mathbf{W}_{L+1}|\{\mathbf{W}_\ell\}_{\ell=1}^L\right)$ for any given $\mathrm{Q}\left(\{\mathbf{W}_\ell\}_{\ell=1}^L\right)$. We begin by simplifying the ELBO by incorporating terms that do not depend on $\mathbf{W}_{L+1}$ into a constant, $c$,

$$\mathcal{L} = \mathbb{E}_{\mathrm{Q}\left(\{\mathbf{W}_\ell\}_{\ell=1}^{L+1}\right)}\left[\log \mathrm{P}\left(\mathbf{Y}, \mathbf{W}_{L+1}|\mathbf{X}, \{\mathbf{W}_\ell\}_{\ell=1}^L\right) - \log \mathrm{Q}\left(\mathbf{W}_{L+1}|\{\mathbf{W}_\ell\}_{\ell=1}^L\right) + c\right]. \qquad (4)$$

Rearranging these terms, we find that all $\mathbf{W}_{L+1}$ dependence can be written in terms of the KL divergence between the approximate posterior of interest and the true posterior,

$$\mathcal{L} = \mathbb{E}_{\mathrm{Q}\left(\{\mathbf{W}_\ell\}_{\ell=1}^L\right)}\big[\log \mathrm{P}\left(\mathbf{Y}|\mathbf{X}, \{\mathbf{W}_\ell\}_{\ell=1}^L\right)$$
$$- \mathrm{D}_{KL}\left(\mathrm{Q}\left(\mathbf{W}_{L+1}|\{\mathbf{W}_\ell\}_{\ell=1}^L\right) \,\|\, \mathrm{P}\left(\mathbf{W}_{L+1}|\mathbf{Y}, \mathbf{X}, \{\mathbf{W}_\ell\}_{\ell=1}^L\right)\right) + c\big]. \qquad (5)$$

Thus, the optimal approximate posterior is,

$$\mathrm{Q}\left(\mathbf{W}_{L+1}|\{\mathbf{W}_\ell\}_{\ell=1}^L\right) = \mathrm{P}\left(\mathbf{W}_{L+1}|\mathbf{Y}, \mathbf{X}, \{\mathbf{W}_\ell\}_{\ell=1}^L\right) \propto \mathrm{P}\left(\mathbf{Y}|\mathbf{W}_{L+1}, \mathbf{F}_L\right)\mathrm{P}\left(\mathbf{W}_{L+1}\right), \qquad (6)$$

and where the final proportionality comes by applying Bayes theorem and exploiting the model's conditional independencies. For regression, the likelihood is Gaussian,

$$\mathrm{P}\left(\mathbf{Y}|\mathbf{W}_{L+1}, \mathbf{F}_L\right) = \textstyle\prod_{\lambda=1}^{N_{L+1}} \mathcal{N}\left(\mathbf{y}_\lambda; \phi\left(\mathbf{F}_L\right)\mathbf{w}_\lambda^{L+1}, \mathbf{\Lambda}_{L+1}^{-1}\right), \qquad (7)$$

where $\mathbf{y}_\lambda$ is the value of a single output channel for all training inputs, and $\mathbf{\Lambda}_{L+1}$ is a precision matrix. Thus, the posterior is given in closed form by Bayesian linear regression (Rasmussen & Williams, 2006).

### 2.1 DEFINING THE FULL APPROXIMATE POSTERIOR WITH GLOBAL INDUCING POINTS AND PSEUDO-DATA

We adapt the optimal scheme above to give a scalable approximate posterior over the weights at all layers. To avoid propagating all training inputs through the network, which is intractable for large datasets, we instead propagate $M$ *global* inducing locations, $\mathbf{U}_0$,

$$\mathbf{U}_1 = \mathbf{U}_0\mathbf{W}_1, \qquad \mathbf{U}_\ell = \phi\left(\mathbf{U}_{\ell-1}\right)\mathbf{W}_\ell \quad \text{for } \ell = 2, \ldots, L+1. \qquad (8)$$

Next, the optimal posterior requires outputs, $\mathbf{Y}$. However, no outputs are available at inducing locations for the output layer, let alone for intermediate layers. We thus introduce learned variational parameters to mimic the form of the optimal posterior. In particular, we use the product of the prior over weights and a "pseudo-likelihood", $\mathcal{N}\left(\mathbf{v}_\lambda^\ell; \mathbf{u}_\lambda^\ell, \mathbf{\Lambda}_\ell^{-1}\right)$, representing noisy "pseudo-observations" of the outputs of the linear layer at the inducing locations, $\mathbf{u}_\lambda^\ell = \phi\left(\mathbf{U}_{\ell-1}\right)\mathbf{w}_\lambda^\ell$. Substituting $\mathbf{u}_\lambda^\ell$ into the pseudo-likelihood the approximate posterior becomes,

$$\mathrm{Q}\left(\mathbf{W}_\ell | \{\mathbf{W}_{\ell'}\}_{\ell'=1}^{\ell-1}\right) \propto \prod_{\lambda=1}^{N_\ell} \mathcal{N}\left(\mathbf{v}_\lambda^\ell; \phi\left(\mathbf{U}_{\ell-1}\right)\mathbf{w}_\lambda^\ell, \mathbf{\Lambda}_\ell^{-1}\right) \mathrm{P}\left(\mathbf{w}_\lambda^\ell\right),$$

$$\mathrm{Q}\left(\mathbf{W}_\ell \Big| \{\mathbf{W}_{\ell'}\}_{\ell'=1}^{\ell-1}\right) = \prod_{\lambda=1}^{N_\ell} \mathcal{N}\left(\mathbf{w}_\lambda^\ell \Big| \mathbf{\Sigma}_\ell^\mathbf{w} \phi\left(\mathbf{U}_{\ell-1}\right)^T \mathbf{\Lambda}_\ell \mathbf{v}_\lambda^\ell, \mathbf{\Sigma}_\ell^\mathbf{w}\right),$$

$$\mathbf{\Sigma}_\ell^\mathbf{w} = \left(N_{\ell-1}\mathbf{S}_\ell^{-1} + \phi\left(\mathbf{U}_{\ell-1}\right)^T \mathbf{\Lambda}_\ell \phi\left(\mathbf{U}_{\ell-1}\right)\right)^{-1}. \tag{9}$$

where $\mathbf{v}_\lambda^\ell$ and $\mathbf{\Lambda}_\ell$ are variational parameters. Therefore, our full approximate posterior factorises as

$$\mathrm{Q}\left(\{\mathbf{W}_\ell\}_{\ell=1}^{L+1}\right) = \prod_{\ell=1}^{L+1} \mathrm{Q}\left(\mathbf{W}_\ell \Big| \{\mathbf{W}_{\ell'}\}_{\ell'=1}^{\ell-1}\right). \tag{10}$$

Thus, the full ELBO can be written,

$$\mathcal{L} = \mathbb{E}_{\mathrm{Q}\left(\{\mathbf{W}\}_{\ell=1}^{L+1}\right)}\left[\log \mathrm{P}\left(\mathbf{Y}|\mathbf{X}, \{\mathbf{W}\}_{\ell=1}^{L+1}\right) + \log \mathrm{P}\left(\{\mathbf{W}_\ell\}_{\ell=1}^{L+1}\right) - \log \mathrm{Q}\left(\{\mathbf{W}_\ell\}_{\ell=1}^{L+1}\right)\right] \tag{11}$$

$$= \mathbb{E}_{\mathrm{Q}\left(\{\mathbf{W}\}_{\ell=1}^{L+1}\right)}\left[\log \mathrm{P}\left(\mathbf{Y}, |\mathbf{X}, \{\mathbf{W}\}_{\ell=1}^{L+1}\right) + \sum_{\ell=1}^{L+1} \log \frac{\mathrm{P}\left(\mathbf{W}_\ell\right)}{\mathrm{Q}\left(\mathbf{W}_\ell | \{\mathbf{W}_\ell\}_{\ell=1}^{\ell-1}\right)}\right]. \tag{12}$$

The forms of the ELBO and approximate posterior suggest a sequential procedure to evaluate and subsequently optimise it: we alternate between sampling the weights using Eq. (9) and propagating the data and inducing locations using Eq. (8) (see Alg. 1). In summary, the parameters of the approximate posterior are the global inducing inputs, $\mathbf{U}_0$, and the pseudo-data and precisions at all layers, $\{\mathbf{V}_\ell, \mathbf{\Lambda}_\ell\}_{\ell=1}^{L+1}$. As each factor $\mathrm{Q}\left(\mathbf{W}_\ell \Big| \{\mathbf{W}_{\ell'}\}_{\ell'=1}^{\ell-1}\right)$ is a Gaussian, these parameters can be optimised using standard reparameterised variational inference (Kingma & Welling, 2013; Rezende et al., 2014) in combination with the Adam optimiser (Kingma & Ba, 2014) (Appendix A). Importantly, by placing inducing inputs on the training data (i.e. $\mathbf{U}_0 = \mathbf{X}$), and setting $\mathbf{v}_\lambda^\ell = \mathbf{y}_\lambda$ this approximate posterior matches the optimal top-layer posterior (Eq. 6). Finally, we note that while this posterior is conditionally Gaussian, the full posterior over all $\{\mathbf{W}_\ell\}_{\ell=1}^{L+1}$ is non-Gaussian, and is thus potentially more flexible than a full-covariance Gaussian over all weights.

---

**Algorithm 1:** Global inducing points for neural networks

---

**Parameters:** inducing inputs, $\mathbf{U}_0$, inducing outputs and precisions, $\{\mathbf{V}_\ell, \mathbf{\Lambda}_\ell\}_{\ell=1}^{L}$, at all layers.
**Neural network inputs:** (e.g. MNIST digits) $\mathbf{F}_0$
**Neural network outputs:** (e.g. classification logits) $\mathbf{F}_{L+1}$
$\mathcal{L} + \infty \leftarrow 0$
**for** $\ell \in \{1, \ldots, L+1\}$ **do**
    *Compute the mean and covariance over the weights at this layer*
    $\mathbf{\Sigma}_\ell^\mathbf{w} = \left(N_{\ell-1}\mathbf{S}_\ell^{-1} + \phi\left(\mathbf{U}_{\ell-1}\right)^T \mathbf{\Lambda}_\ell \phi\left(\mathbf{U}_{\ell-1}\right)\right)^{-1}$
    $\mathbf{M}_\ell = \mathbf{\Sigma}_\ell^\mathbf{w} \phi\left(\mathbf{U}_{\ell-1}\right)^T \mathbf{\Lambda}_\ell \mathbf{V}_\ell$
    *Sample the weights and compute the ELBO*
    $\mathbf{W}_\ell \sim \mathcal{N}\left(\mathbf{M}_\ell, \mathbf{\Sigma}_\ell^\mathbf{w}\right) = \mathrm{Q}\left(\mathbf{W}_\ell \Big| \{\mathbf{W}_{\ell'}\}_{\ell'=1}^{\ell-1}\right)$
    $\mathcal{L} \leftarrow \mathcal{L} + \log \mathrm{P}\left(\mathbf{W}_\ell\right) - \log \mathcal{N}\left(\mathbf{W}_\ell | \mathbf{M}_\ell, \mathbf{\Sigma}_\ell^\mathbf{w}\right)$
    *Propagate the inputs and inducing points using the sampled weights,*
    $\mathbf{U}_\ell = \phi\left(\mathbf{U}_{\ell-1}\right)\mathbf{W}_\ell$
    $\mathbf{F}_\ell = \phi\left(\mathbf{F}_{\ell-1}\right)\mathbf{W}_\ell$
$\mathcal{L} \leftarrow \mathcal{L} + \log \mathrm{P}\left(\mathbf{Y}|\mathbf{F}_{L+1}\right)$

---

## 2.2 Efficient convolutional Bayesian linear regression

The previous sections were valid for a fully connected network. The extension to convolutional networks is straightforward in principle: we transform the convolution into a matrix multiplication by treating each patch as a separate input feature-vector, flattening the spatial and channel dimensions together into a single vector. Thus, the feature-vectors have length `in_channels` $\times$ `kernel_width` $\times$ `kernel_height`, and the matrix $\mathbf{U}_\ell$ contains `patches_per_image` $\times$ `minibatch` patches. Likewise, we now have inducing outputs, $\mathbf{v}_\lambda^\ell$, at each location in all the inducing images, so this again has length `patches_per_image` $\times$ `minibatch`. After explicitly extracting the patches, we can straightforwardly apply standard Bayesian linear regression.

However, explicitly extracting image patches is very memory intensive in a DNN. If we consider a standard convolution with a $3 \times 3$ convolutional kernel, then there is a $3 \times 3$ patch centred at each pixel in the input image, meaning a factor of 9 increase in memory consumption. Instead, we note that computing the matrices required for linear regression, $\phi\left(\mathbf{U}_{\ell-1}\right)^T \boldsymbol{\Lambda}_\ell \phi\left(\mathbf{U}_{\ell-1}\right)$ and $\phi\left(\mathbf{U}_\ell\right)^T \boldsymbol{\Lambda}_\ell \mathbf{V}_\ell$, does not require explicit extraction of image-patches. Instead, these matrices can be computed by taking the autocorrelation of the image/feature map, i.e. a convolution operation where we treat the image/feature map, as *both* the inputs and the weights (Appendix B for details).

## 2.3 Priors

We consider four priors in this work, which we refer to using the class names in the BNN library published with this paper. We are careful to ensure that all parameters in the model have a prior and approximate posterior, which is necessary to ensure that ELBOs are comparable across models.

First, we consider a Gaussian prior with fixed scale, NealPrior, so named because it is necessary to obtain meaningful results when considering infinite networks (Neal, 1996),

$$\mathbf{S}_\ell = \mathbf{I}, \tag{13}$$

though it bears strong similarities to the "He" initialisation (He et al., 2015). NealPrior is defined so as to ensure that the activations retain a sensible scale as they propagate through the network. We compare this to the standard $\mathcal{N}(0, 1)$ (StandardPrior), which causes the activations to increase exponentially as they propagate through network layers (see Eq. 2):

$$\mathbf{S}_\ell = N_{\ell-1}\mathbf{I}. \tag{14}$$

Next, we consider ScalePrior, which defines a prior and approximate posterior over the scale,

$$\mathbf{S}_\ell = \frac{1}{s_\ell}\mathbf{I} \qquad \mathrm{P}\left(s_\ell\right) = \mathrm{Gamma}\left(s_\ell; 2, 2\right) \qquad \mathrm{Q}\left(s_\ell\right) = \mathrm{Gamma}\left(s_\ell; 2 + \alpha_\ell, 2 + \beta_\ell\right) \tag{15}$$

where here we parameterise the Gamma distribution with the shape and rate parameters, and $\alpha_\ell$ and $\beta_\ell$ are non-negative learned parameters of the approximate posterior over $s_\ell$. Finally, we consider SpatialIWPrior, which allows for spatial correlations in the weights. In particular, we take the covariance to be the Kronecker product of an identity matrix over channel dimensions, and a Wishart-distributed matrix, $\mathbf{L}_\ell^{-1}$, over the spatial dimensions,

$$\mathbf{S}_\ell = \mathbf{I} \otimes \mathbf{L}_\ell^{-1}$$
$$\mathrm{P}\left(\mathbf{L}_\ell\right) = \mathrm{InverseWishart}\left(\mathbf{L}_\ell; \left(N_{\ell-1} + 1\right)\mathbf{I} \qquad , N_{\ell-1} + 1\right)$$
$$\mathrm{Q}\left(\mathbf{L}_\ell\right) = \mathrm{InverseWishart}\left(\mathbf{L}_\ell; \left(N_{\ell-1} + 1\right)\mathbf{I} + \boldsymbol{\Psi}, N_{\ell-1} + 1 + \nu\right) \tag{16}$$

where the non-negative real number, $\nu$, and the positive definite matrix, $\boldsymbol{\Psi}$, are learned parameters of the approximate posterior (see Appendix C).

## 3 Extension to DGPs

It is a remarkable but underappreciated fact that BNNs are special cases of DGPs, with a particular choice of kernel (Louizos & Welling, 2016; Aitchison, 2019). Combining Eqs. (1) and (2),

$$\mathrm{P}\left(\mathbf{F}_\ell | \mathbf{F}_{\ell-1}\right) = \prod_{\lambda=1}^{N_\ell} \mathcal{N}\left(\mathbf{f}_\lambda^\ell | \mathbf{0}, \mathbf{K}\left(\mathbf{F}_{\ell-1}\right)\right), \quad \mathbf{K}\left(\mathbf{F}_{\ell-1}\right) = \frac{1}{N_{\ell-1}}\phi\left(\mathbf{F}_{\ell-1}\right)\mathbf{S}_\ell \phi\left(\mathbf{F}_{\ell-1}\right)^T. \tag{17}$$

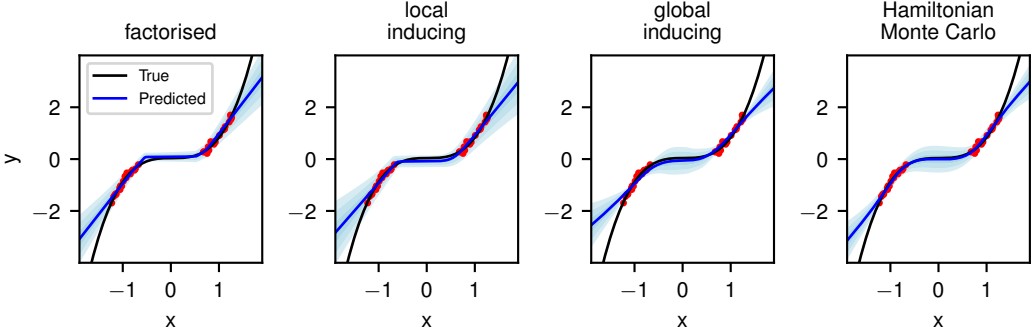

Figure 1: Predictive distributions on the toy dataset. Shaded regions represent one standard deviation.

Here, we generalise our approximate posterior to the DGP case and link to the DGP literature. In a DGP there are no weights; instead we work directly with inducing outputs $\{\mathbf{U}_\ell\}_{\ell=1}^{L+1}$,

$$\mathrm{P}\left(\mathbf{U}_\ell|\mathbf{U}_{\ell-1}\right) = \prod_{\lambda=1}^{N_\ell}\mathcal{N}\left(\mathbf{u}_\lambda^\ell|\mathbf{0},\mathbf{K}\left(\mathbf{U}_{\ell-1}\right)\right), \tag{18}$$

Note that here, we take the "global" inducing approach of using the inducing outputs from the previous layer, $\mathbf{U}_{\ell-1}$ as the inducing inputs for the next layer. In this case, we need only learn the original inducing inputs, $\mathbf{U}_0$. This contrasts with the standard "local" inducing formulation, (as in Salimbeni & Deisenroth, 2017), which learns separate inducing inputs at every layer, $\mathbf{Z}_{\ell-1}$, giving $\mathrm{P}\left(\mathbf{U}_\ell|\mathbf{Z}_{\ell-1}\right) = \prod_{\lambda=1}^{N_\ell}\mathcal{N}\left(\mathbf{u}_\lambda^\ell|\mathbf{0},\mathbf{K}\left(\mathbf{Z}_{\ell-1}\right)\right)$.

As usual in DGPs (Salimbeni & Deisenroth, 2017), the approximate posterior over $\mathbf{U}_\ell$ induces an approximate posterior on $\mathbf{F}_\ell$ through the prior correlations. However, it is important to remember that underneath the tractable distributions in Eqs. (17) and (18), there is an infinite dimensional GP-distributed function, $\mathcal{F}_\ell$, such that $\mathbf{F}_\ell = \mathcal{F}_\ell(\mathbf{F}_{\ell-1})$. Standard local inducing point methods specify a factorised approximate posterior over $\mathcal{F}_\ell$ by specifying the function's inducing outputs, $\mathbf{U}_\ell = \mathcal{F}_\ell(\mathbf{Z}_{\ell-1})$, at a finite number of inducing input locations, $\mathbf{Z}_{\ell-1}$. Importantly, the approximate posterior over a function, $\mathcal{F}_\ell$, depends only on $\mathbf{Z}_{\ell-1}$, and $\mathbf{U}_\ell$. Thus, standard, local inducing, DGP approaches (e.g. Salimbeni & Deisenroth, 2017), give a layerwise-independent approximate posterior over $\{\mathcal{F}_\ell\}_{\ell=1}^{L+1}$, as they treat the inducing inputs, $\{\mathbf{Z}_{\ell-1}\}_{\ell=1}^{L+1}$, as fixed, learned parameters and use a layerwise-independent approximate posterior over $\{\mathbf{U}_\ell\}_{\ell=1}^{L+1}$ (Appendix F).

Next, we need to choose the approximate posterior on $\{\mathbf{U}_\ell\}_{\ell=1}^{L+1}$. However, if our goal is to introduce dependence across layers, it seems inappropriate to use the standard layerwise-independent approximate posterior over $\{\mathbf{U}_\ell\}_{\ell=1}^{L+1}$. Indeed, in Appendix F, we show that such a posterior implies functions in non-adjacent layers (e.g. $\mathcal{F}_\ell$ and $\mathcal{F}_{\ell+2}$) are marginally independent, even with global inducing points.

To obtain more appropriate approximate posteriors, we derive the optimal top-layer posterior for DGPs, which involves GP regression from activations propagated from lower layers onto the output data (Appendix E). Inspired by the form of the optimal posterior we again define an approximate posterior by taking the product of the prior and a "pseudo-likelihood",

$$\mathrm{Q}\left(\mathbf{U}_\ell|\mathbf{U}_{\ell-1}\right) \propto \prod_{\lambda=1}^{N_\ell}\mathrm{Q}\left(\mathbf{u}_\lambda^\ell\right)\mathcal{N}\left(\mathbf{v}_\lambda^\ell|\mathbf{u}_\lambda^\ell,\mathbf{\Lambda}_\ell^{-1}\right)$$

$$\mathrm{Q}\left(\mathbf{U}_\ell|\mathbf{U}_{\ell-1}\right) = \prod_{\lambda=1}^{N_\ell}\mathcal{N}\left(\mathbf{u}_\lambda^\ell|\mathbf{\Sigma}_\ell^{\mathbf{u}}\mathbf{\Lambda}_\ell\mathbf{v}_\lambda^\ell,\mathbf{\Sigma}_\ell^{\mathbf{u}}\right), \qquad \mathbf{\Sigma}_\ell^{\mathbf{u}} = \left(\mathbf{K}^{-1}\left(\mathbf{U}_{\ell-1}\right)+\mathbf{\Lambda}_\ell\right)^{-1}, \tag{19}$$

where $\mathbf{v}_\lambda^\ell$ and $\mathbf{\Lambda}_\ell^{-1}$ are learned parameters, and in our global inducing method, the inducing inputs, $\mathbf{U}_{\ell-1}$, are propagated from lower layers (Eq. 18). Importantly, setting the inducing inputs to the training data and $\mathbf{v}_\lambda^{L+1} = \mathbf{y}_\lambda$, the approximate posterior captures the optimal top-layer posterior for regression (Appendix E). Under this approximate posterior, dependencies in $\mathbf{U}_\ell$ naturally arise across all layers, and hence there are dependencies between functions $\mathcal{F}_\ell$ at all layers (Appendix F).

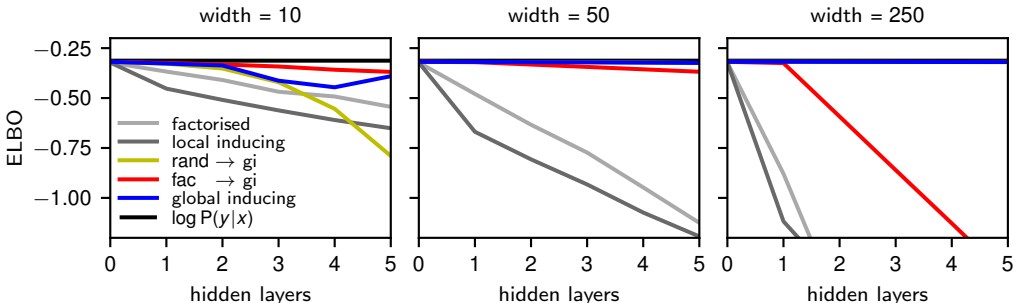

Figure 2: ELBO for different approximate posteriors as we change network depth/width on a dataset generated using a linear Gaussian model. The rand $\to$ gi line lies behind the global inducing line in width $= 50$ and width $= 250$.

In summary, we propose an approximate posterior over inducing outputs that takes the form

$$Q\left(\{\mathbf{U}_\ell\}_{\ell=1}^{L+1}\right) = \prod_{l=1}^{L+1} Q\left(\mathbf{U}_\ell | \mathbf{U}_{\ell-1}\right). \tag{20}$$

As before, the parameters of this approximate posterior are the global inducing inputs, $\mathbf{U}_0$, and the pseudo-data and precisions at all layers, $\{\mathbf{V}_\ell, \boldsymbol{\Lambda}_\ell\}_{\ell=1}^{L+1}$. Our ELBO, which we derive in D, takes the form

$$\mathcal{L} = \mathbb{E}_{Q\left(\{\mathbf{F}_\ell, \mathbf{U}_\ell\}_{\ell=1}^{L+1} | \mathbf{X}, \mathbf{U}_0\right)} \left[ \log P\left(\mathbf{Y} | \mathbf{F}_{L+1}\right) + \sum_{\ell=1}^{L+1} \log \frac{P\left(\mathbf{U}_\ell | \mathbf{U}_{\ell-1}\right)}{Q\left(\mathbf{U}_\ell | \mathbf{U}_{\ell-1}\right)} \right]. \tag{21}$$

We provide a full description of our method as applied to DGPs in Appendix D. In concluding our discussion of our proposed method, we note that we provide an analysis of the asymptotic complexity in Appendix L.

## 4 RESULTS

We describe our experiments and results to assess the performance of global inducing points ('gi') against local inducing points ('li') and the fully factorised ('fac') approximation family. We additionally consider models where we use one method up to the last layer and another for the last layer, which may have computational advantages; we denote such models 'method1 $\to$ method2'.

### 4.1 UNCERTAINTY IN 1D REGRESSION

We demonstrate the use of local and global inducing point methods in a toy 1-D regression problem, comparing it with fully factorised VI and Hamiltonian Monte Carlo (HMC; (Neal et al., 2011)). Following Hernández-Lobato & Adams (2015), we generate 40 input-output pairs $(x, y)$ with the inputs $x$ sampled i.i.d. from $\mathcal{U}([-4, -2] \cup [2, 4])$ and the outputs generated by $y = x^3 + \epsilon$, where $\epsilon \sim \mathcal{N}(0, 3^2)$. We then normalised the inputs and outputs. Note that we have introduced a 'gap' in the inputs, following recent work (Foong et al., 2019b; Yao et al., 2019; Foong et al., 2019a) that identifies the ability to express 'in-between' uncertainty as an important quality of approximate inference algorithms. We evaluated the inference algorithms using fully-connected BNNs with 2 hidden layers of 50 ReLU hidden units, using the NealPrior. For the inducing point methods, we used 100 inducing points per layer.

The predictive distributions for the toy experiment can be seen in Fig. 1. We observe that of the variational methods, the global inducing method produces predictive distributions closest to HMC, with good uncertainty in the gap. Meanwhile, factorised and local inducing fit the training data, but do not produce reasonable error bars: this demonstrates an important limitation of methods lacking correlation structure between layers.

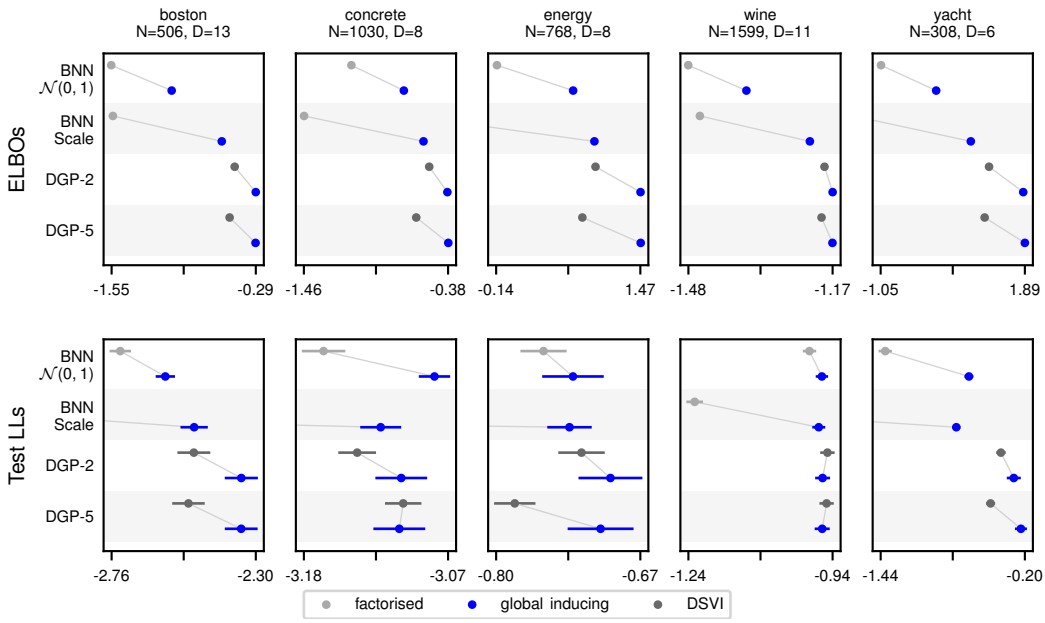

Figure 3: Average test log likelihoods for BNNs on the UCI datasets (in nats). Error bars represent one standard error. Shading represents different priors. We connect the factorised models with the fac $\rightarrow$ gi models with a thin grey line as an aid for easier comparison. Further to the right is better.

We provide additional experiments looking at the effect of the number of inducing points in Appendix H, and experiments looking at compositional uncertainty (Ustyuzhaninov et al., 2020) in both BNNs and DGPs for 1D regression in Appendix I.

## 4.2 DEPTH-DEPENDENCE IN DEEP LINEAR NETWORKS

The lack of correlations between layers might be expected to become more problematic in deeper networks. To isolate the effect of depth on different approximate posteriors, we considered deep linear networks trained on data generated from a toy linear model: 5 input features were mapped to 1 output feature, where the 1000 training and 100 test inputs are drawn IID from a standard Gaussian, and the true outputs are drawn using a weight-vector drawn IID from a Gaussian with variance $1/5$, and with noise variance of $0.1$. We could evaluate the model evidence under the true data generating process which forms an upper bound (in expectation) on the model evidence and ELBO for all models.

We found that the ELBO for methods that factorise across layers — factorised and local inducing — drops rapidly as networks get deeper and wider (Fig. 2). This is undesirable behaviour, as we know that wide, deep networks are necessary for good performance on difficult machine learning tasks. In contrast, we found that methods with global inducing points at the last layer decay much more slowly with depth, and perform better as networks get wider. Remarkably, global-inducing points gave good performance even with lower-layer weights drawn at random from the prior, which is not possible for any method that factorises across layers. We believe that fac $\rightarrow$ gi performed poorly at width $= 250$ due to optimisation issues as rand $\rightarrow$ gi performs better yet is a special case of fac $\rightarrow$ gi.

## 4.3 REGRESSION BENCHMARK: UCI

We benchmark our methods on the UCI datasets in Hernández-Lobato & Adams (2015), popular benchmark regression datasets for BNNs and DGPs. Following the standard approach (Gal & Ghahramani, 2015), each dataset uses 20 train-test 'splits' (except for protein with 5 splits) and the inputs and outputs are normalised to have zero mean and unit standard deviation. We focus on the five smallest datasets, as we expect Bayesian methods to be most relevant in small-data settings (see App. J and M for all datasets). We consider two-layer fully-connected ReLU networks, using fully factorised and global inducing approximating families, as well as two- and five-layer DGPs with

Table 1: CIFAR-10 classification accuracy. The first block shows our results using SpatialIWPrior, with ScalePrior in brackets. The next block shows comparable past results, from GPs and BNNs. The final block show non-comparable (sampling-based) methods. Dashes indicate that the figures were either not reported, are not applicable. The time is reported per epoch with ScalePrior and for MNIST, rather than CIFAR-10 because of a known performance bug in the convolutions required in Sec. 2.2 with $32 \times 32$ (and above) images `https://github.com/pytorch/pytorch/issues/35603`.

|  | test log like. | accuracy (%) | ELBO | time |
|---|---|---|---|---|
| factorised | -0.58 (-0.66) | 80.27 (77.65) | -1.06 (-1.12) | 19 s |
| local inducing | -0.62 (-0.60) | 78.96 (79.46) | -0.84 (-0.88) | 33 s |
| fac $\to$ gi | -0.49 (-0.56) | 83.33 (81.72) | -0.91 (-0.96) | 25 s |
| **global inducing** | **-0.40 (-0.43)** | **86.70 (85.73)** | **-0.68 (-0.75)** | 65 s |
| Shi et al. (2019) | — | 80.30% | — |  |
| Li et al. (2019) | — | 81.40% | — |  |
| Shridhar et al. (2019) | — | 73% | — |  |
| Wenzel et al. (2020) | $-0.35$ | 88.50% | — |  |

doubly-stochastic variational inference (DSVI) (Salimbeni & Deisenroth, 2017) and global inducing. For the BNNs, we consider the standard $\mathcal{N}(0, 1)$ prior and ScalePrior.

We display ELBOs and average test log likelihoods for the un-normalised data in Fig. 3, where the dots and error bars represent the means and standard errors over the test splits, respectively. We observe that global inducing obtains better ELBOs than factorised and DSVI in every case, indicating that it does indeed approximate the true posterior better (since the ELBO is the marginal likelihood minus the KL to the posterior). While this is the case for the ELBOs, this does not always translate to a better test log likelihood due to model misspecification, as we see that occasionally DSVI outperforms global inducing by a very small margin. The very poor results for factorised on ScalePrior indicate that it has difficulty learning useful prior hyperparameters for prediction, which is due to the looseness of its bound to the marginal likelihood. We provide experimental details, as well as additional results with additional architectures, priors, datasets, and RMSEs, in Appendices J and M, for BNNs and DGPs, respectively.

## 4.4 CONVOLUTIONAL BENCHMARK: CIFAR-10

For CIFAR-10, we considered a ResNet-inspired model consisting of conv2d-relu-block-avgpool2-block-avgpool2-block-avgpool-linear, where the ResNet blocks consisted of a shortcut connection in parallel with conv2d-relu-conv2d-relu, using 32 channels in all layers. In all our experiments, we used no data augmentation and 500 inducing points. Our training scheme (see App. O) ensured that our results did not reflect a 'cold posterior' (Wenzel et al., 2020). Our results are shown in Table 1. We achieved remarkable performance of $86.7\%$ predictive accuracy, with global inducing points used for all layers, and with a spatial inverse Wishart prior on the weights. These results compare very favourably with comparable Bayesian approaches, i.e. those without data augmentation or posterior sharpening: past work with deep GPs obtained $80.3\%$ (Shi et al., 2019), and work using infinite-width neural networks to define a GP obtained $81.4\%$ accuracy (Li et al., 2019). Remarkably, with only 500 inducing points we are approaching the accuracy of sampling-based methods (Wenzel et al., 2020), which are in principle able to more closely approximate the true posterior. Furthermore, we see that global inducing performs the best in terms of ELBO (per datapoint) by a wide margin, demonstrating that it gets far closer to the true posterior than the other methods. We provide additional results on uncertainty calibration in Appendix K.

## 5 RELATED WORK

Louizos & Welling (2016) attempted to use pseudo-data along with matrix variate Gaussians to form an approximate posterior for BNNs; however, they restricted their analysis to BNNs, and it is not clear how their method can be applied to DGPs. Their approach factorises across layers, thus

missing the important layerwise correlations that we obtain. Moreover, they encountered an important limitation: the BNN prior implies that $\mathbf{U}_\ell$ is low-rank and it is difficult to design an approximate posterior capturing this constraint. As such, they were forced to use $M < N_\ell$ inducing points, which is particularly problematic in the convolutional, global-inducing case where there are many patches (points) in each inducing image input.

Note that some work on BNNs reports better performance on datasets such as CIFAR-10. However, to the best of our knowledge, no variational Bayesian method outperforms ours without modifying the BNN model or some form of posterior tempering (Wenzel et al., 2020), where the KL term in the ELBO is down-weighted relative to the likelihood (Zhang et al., 2017; Bae et al., 2018; Osawa et al., 2019; Ashukha et al., 2020), which often increases the test accuracy. However, tempering clouds the Bayesian perspective, as the KL to the posterior is no longer minimised and the resulting objective is no longer a lower bound on the marginal likelihood. By contrast, we use the untempered ELBO, thereby retaining the full Bayesian perspective. Dusenberry et al. (2020) report better performance on CIFAR-10 without tempering, but only perform variational inference over a rank-1 perturbation to the weights, and maximise over all the other parameters. Our approach retains a full-rank parameterisation of the weight matrices.

Ustyuzhaninov et al. (2020) attempted to introduce dependence across layers in a deep GP by coupling inducing inputs to pseudo-outputs, which they term "inducing points as inducing locations". However, as described above, the choice of approximate posterior over $\mathbf{U}_\ell$ is also critical. They used the standard approximate posterior that is independent across layers, meaning that while functions in adjacent layers were marginally dependent, the functions for non-adjacent layers were independent (Appendix F). By contrast, our approximate posteriors have marginal dependencies across $\mathbf{U}_\ell$ and functions at all layers, and are capable of capturing the optimal top-layer posterior.

## 6 CONCLUSIONS

We derived optimal top-layer variational approximate posteriors for BNNs and deep GPs, and used them to develop generic, scalable approximate posteriors. These posteriors make use of *global* inducing points, which are learned only at the bottom layer and are propagated through the network. This leads to extremely flexible posteriors, which even allow the lower-layer weights to be drawn from the prior. We showed that these global inducing variational posteriors lead to improved performance with better ELBOs, and state-of-the-art performance for variational BNNs on CIFAR-10.

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

## A    REPARAMETERISED VARIATIONAL INFERENCE

In variational inference, the ELBO objective takes the form,

$$\mathcal{L}(\phi) = \mathbb{E}_{Q_\phi(\mathbf{w})} \left[ \log P\left(\mathbf{Y}|\mathbf{X}, \mathbf{w}\right) + \log \frac{P\left(\mathbf{w}\right)}{Q_\phi\left(\mathbf{w}\right)} \right] \tag{22}$$

where $\mathbf{w}$ is a vector containing all of the elements of the weight matrices in the full network, $\{\mathbf{W}_\ell\}_{\ell=1}^{L+1}$, and $\phi = (\mathbf{U}_0, \{\mathbf{V}_\ell, \mathbf{\Lambda}_\ell\}_{\ell=1}^{L+1})$ are the parameters of the approximate posterior. This objective is difficult to differentiate wrt $\phi$, because $\phi$ parameterises the distribution over which the expectation is taken. Following Kingma & Welling (2013) and Rezende et al. (2014), we sample $\epsilon$ from a simple, fixed distribution (e.g. IID Gaussian), and transform them to give samples from $Q(w)$,

$$\mathbf{w}(\epsilon; \phi) \sim Q_\phi\left(\mathbf{w}\right). \tag{23}$$

Thus, the ELBO can be written,

$$\mathcal{L}(\phi) = \mathbb{E}_\epsilon \left[ \log P\left(\mathbf{Y}|\mathbf{X}, \mathbf{w}(\epsilon; \phi)\right) + \log \frac{P\left(\mathbf{w}(\epsilon; \phi)\right)}{Q_\phi\left(\mathbf{w}(\epsilon; \phi)\right)} \right] \tag{24}$$

As the distribution over which the expectation is taken is now independent of $\phi$, we can form unbiased estimates of the gradient of $\mathcal{L}(\phi)$ by drawing one or a few samples of $\epsilon$.

## B    EFFICIENT CONVOLUTIONAL LINEAR REGRESSION

Working in one dimension for simplicity, the standard form for a convolution in deep learning is

$$Y_{iu,c} = \sum_{c'\delta} X_{i(u+\delta)c'} W_{\delta c',c}. \tag{25}$$

where $X$ is the input image/feature-map, $Y$ is the output feature-map, $W$ is the convolutional weights, $i$ indexes images, $c$ and $c'$ index channels, $u$ indexes the location within the image, and $\delta$ indexes the location within the convolutional patch. Later, we will swap the identity of the "patch location" and the "image location" and to facilitate this, we define them both to be centred on zero,

$$u \in \{-(W-1)/2, \ldots, (W-1)/2\} \qquad \delta \in \{-(S-1)/2, \ldots, (S-1)/2\} \tag{26}$$

where $W$ is the size of an image and $S$ is the size of a patch, such that, for example for a size 3 kernel, $\delta \in \{-1, 0, 1\}$.

To use the expressions for the fully-connected case, we can form a new input, $X'$ by cutting out each image patch,

$$X'_{iu,\delta c} = X_{i(u+\delta)c}, \tag{27}$$

leading to

$$Y_{iu,c} = \sum_{\delta c'} X'_{iu,\delta c'} W_{\delta c',c}. \tag{28}$$

Note that we have used commas to group pairs of indices ($iu$ and $\delta c'$) that may be combined into a single index (e.g. using a reshape operation). Indeed, combining $i$ and $u$ into a single index and combining $\delta$ and $c'$ into a single index, this expression can be viewed as standard matrix multiplication,

$$\mathbf{Y} = \mathbf{X}'\mathbf{W}. \tag{29}$$

This means that we can directly apply the approximate posterior we derived for the fully-connected case in Eq. (9) to the convolutional case. To allow for this, we take

$$\mathbf{X}' = \mathbf{\Lambda}_\ell^{1/2} \phi\left(\mathbf{U}_{\ell-1}\right), \qquad\qquad \mathbf{Y} = \mathbf{\Lambda}_\ell^{1/2} \mathbf{V}_\ell. \tag{30}$$

While we can explicitly compute $\mathbf{X}'$ by extracting image patches, this imposes a very large memory cost (a factor of 9 for a $3 \times 3$ kernel, with stride 1, because there are roughly as many patches as pixels in the image, and a $3 \times 3$ patch requires 9 times the storage of a pixel. To implement convolutional linear regression with a more manageable memory cost, we instead compute the matrices required

for linear regression directly as convolutions of the input feature-maps, $\mathbf{X}$ with themselves, and as convolutions of the $X$ with the output feature maps, $\mathbf{Y}$, which we describe here.

For linear regression (Eq. 9), we first need to compute,

$$\left(\phi\left(\mathbf{U}_{\ell-1}\right)^T \mathbf{\Lambda}_\ell \mathbf{V}_\ell\right)_{\delta c, c'} = \left(\mathbf{X}'^T \mathbf{Y}\right)_{\delta c, c'} = \sum_{iu} X'_{iu, \delta c} Y_{iu, c'}. \tag{31}$$

Rewriting this in terms of $X$ (i.e. without explicitly cutting out image patches), we obtain,

$$\left(\mathbf{X}'^T \mathbf{Y}\right)_{\delta c, c'} = \sum_{iu} X_{i(u+\delta)c} Y_{iu, c'}. \tag{32}$$

This can be directly viewed as the convolution of $X$ and $Y$, where we treat $Y$ as the "convolutional weights", $u$ as the location within the now very large (size $W$) "convolutional patch", and $\delta$ as the location in the resulting output. Once we realise that the computation is a spatial convolution, it is possible to fit it into standard convolution functions provided by deep-learning frameworks (albeit with some rearrangement of the tensors).

Next, we need to compute,

$$\left(\phi\left(\mathbf{U}_{\ell-1}\right)^T \mathbf{\Lambda}_\ell \phi\left(\mathbf{U}_{\ell-1}\right)\right) = \left(\mathbf{X}'^T \mathbf{X}'\right)_{\delta c, \delta' c'} = \sum_{iu} X'_{iu, \delta c} X'_{iu, \delta' c'}. \tag{33}$$

Again, rewriting this in terms of $X$ (i.e. without explicitly cutting out image patches), we obtain,

$$\left(\mathbf{X}'^T \mathbf{X}'\right)_{\delta c, \delta' c'} = \sum_{iu} X_{i(u+\delta)c} X_{i(u+\delta')c'}. \tag{34}$$

To treat this as a convolution, we first need exact translational invariance, which can be achieved by using circular boundary conditions. Note that circular boundary conditions are not typically used in neural networks for images, and we therefore only use circular boundary conditions to define the approximate posterior over weights. The variational framework does not restrict us to also using circular boundary conditions within our feedforward network, and as such, we use standard zero-padding. With exact translational invariance, we can write this expression directly as a convolution,

$$\left(\mathbf{X}'^T \mathbf{X}'\right)_{\delta c, \delta' c'} = \sum_{iu} X_{iuc} X_{i(u+\delta'-\delta)c'} \tag{35}$$

where

$$(\delta' - \delta) \in \{-(S-1), \dots, (S-1)\} \tag{36}$$

i.e. for a size 3 kernel, $(\delta' - \delta) \in \{-2, -1, 0, 1, 2\}$, where we treat $X_{iuc}$ as the "convolutional weights", $u$ as the location within the "convolutional patch", and $\delta' - \delta$ as the location in the resulting output "feature-map".

Finally, note that this form offers considerable benefits in terms of memory consumption. In particular, the output matrices are usually quite small — the number of channels is typically 32 or 64, and the number of locations within a patch is typically 9, giving a very manageable total size that is typically smaller than $1000 \times 1000$.

## C  Wishart distributions with real-valued degrees of freedom

The classical description of the Wishart distribution,

$$\mathbf{\Sigma} \sim \text{Wishart}\left(\mathbf{I}, \nu\right), \tag{37}$$

where $\mathbf{\Sigma}$ is a $P \times P$ matrix, states that $P \geq \nu$ is an integer and we can generate $\mathbf{\Sigma}$ by taking the product of matrices, $\mathbf{X} \in \mathbb{R}^{\nu \times P}$, generated IID from a standard Gaussian,

$$\mathbf{\Sigma} = \mathbf{X}^T \mathbf{X}, \qquad\qquad X_{ij} \sim \mathcal{N}\left(0, 1\right). \tag{38}$$

However, for the purposes of defining learnable approximate posteriors, we need to be able to sample and evaluate the probability density when $\nu$ is positive real.

To do this, consider the alternative, much more efficient means of sampling from a Wishart distribution, using the Bartlett decomposition (Bartlett, 1933). The Bartlett decomposition gives the probability density for the Cholesky of a Wishart sample. In particular,

$$\mathbf{T} = \begin{pmatrix} T_{11} & \cdots & T_{1m} \\ \vdots & \ddots & \vdots \\ 0 & \cdots & T_{mm} \end{pmatrix}, \tag{39}$$

$$\mathrm{P}\left(T_{jj}^2\right) = \mathrm{Gamma}\left(T_{jj}^2; \tfrac{\nu-j+1}{2}, \tfrac{1}{2}\right), \tag{40}$$

$$\mathrm{P}\left(T_{j<k}\right) = \mathcal{N}\left(T_{jk}; 0, 1\right). \tag{41}$$

Here, $\mathbf{T}_{jj}$ is usually considered to be sampled from a $\chi^2$ distribution, but we have generalised this slightly using the equivalent Gamma distribution to allow for real-valued $\nu$. Following Chafaï (2015), We need to change variables to $T_{jj}$ rather than $T_{jj}^2$,

$$\mathrm{P}\left(T_{jj}\right) = \mathrm{P}\left(T_{jj}^2\right)\left|\frac{\partial T_{jj}^2}{\partial T_{jj}}\right|, \tag{42}$$

$$= \mathrm{Gamma}\left(T_{jj}^2; \tfrac{\nu-j+1}{2}, \tfrac{1}{2}\right) 2T_{jj}, \tag{43}$$

$$= \frac{\left(T_{jj}^2\right)^{(\nu-j+1)/2-1} e^{-T_{jj}^2/2}}{2^{(\nu-j+1)/2}\Gamma\left(\frac{\nu-j+1}{2}\right)} 2T_{jj}, \tag{44}$$

$$= \frac{T_{jj}^{\nu-j} e^{-T_{jj}^2/2}}{2^{(\nu-j-1)/2}\Gamma\left(\frac{\nu-j+1}{2}\right)}. \tag{45}$$

Thus, the probability density for $\mathbf{T}$ under the Bartlett sampling operation is

$$\mathrm{P}\left(\mathbf{T}\right) = \underbrace{\prod_j \frac{T_{jj}^{\nu-j} e^{-T_{jj}^2/2}}{2^{\frac{\nu-j-1}{2}}\Gamma\left(\frac{\nu-j+1}{2}\right)}}_{\text{on-diagonals}} \underbrace{\prod_{k\in\{j+1,\ldots,m\}} \frac{1}{\sqrt{2\pi}} e^{-T_{jk}^2/2}}_{\text{off-diagonals}}. \tag{46}$$

$$\tag{47}$$

To convert this to a distribution on $\boldsymbol{\Sigma}$, we need the volume element for the transformation from $\mathbf{T}$ to $\boldsymbol{\Sigma}$,

$$\mathrm{d}\boldsymbol{\Sigma} = 2^m \prod_{j=1}^m T_{jj}^{m-j+1}\mathrm{d}\mathbf{T}, \tag{48}$$

which can be obtained directly by computing the log-determinant of the Jacobian for the transformation from $\mathbf{T}$ to $\boldsymbol{\Sigma}$, or by taking the ratio of Eq. (46) and the usual Wishart probability density (with integral $\nu$). Thus,

$$\mathrm{P}\left(\boldsymbol{\Sigma}\right) = \mathrm{P}\left(\mathbf{T}\right)\left(2^m \prod_{j=1}^m T_{jj}^{m-j+1}\right)^{-1} \tag{49}$$

$$= \prod_j \frac{T_{jj}^{\nu-m-1} e^{-T_{jj}^2/2}}{2^{\frac{\nu-j+1}{2}}\Gamma\left(\frac{\nu-j+1}{2}\right)} \cdot \prod_{k\in\{j+1,\ldots,m\}} \frac{1}{\sqrt{2\pi}} e^{-T_{jk}^2/2}. \tag{50}$$

Breaking this down into separate components and performing straightforward algebraic manipulations,

$$\prod_j T_{jj}^{\nu-m-1} = |\mathbf{T}|^{\nu-m-1} = |\boldsymbol{\Sigma}|^{(\nu-m-1)/2}, \tag{51}$$

$$\mathrm{P}\left(\boldsymbol{\Sigma}\right) = \prod_j e^{-T_{jj}^2/2} \prod_{k \in \{j+1,\dots,m\}} e^{-T_{jk}^2/2} \tag{52}$$

$$= e^{-\sum_{jk} T_{jk}^2/2} = e^{-\operatorname{Tr}(\boldsymbol{\Sigma})/2}, \tag{53}$$

$$\prod_j \frac{1}{2^{(\nu-j-1)/2}} \prod_{k \in \{j+1,\dots,m\}} \frac{1}{\sqrt{2}} = \left(\prod_j \frac{1}{2^{(\nu-j-1)/2}}\right)\left(\prod_j \frac{1}{2^{(m-j-1)/2}}\right) \tag{54}$$

$$= \left(\prod_j \frac{1}{2^{(\nu-j+1)/2}}\right)\left(\prod_j \frac{1}{2^{(j-1)/2}}\right) \tag{55}$$

$$= \prod_j \frac{1}{2^{(\nu-m)/2}} = 2^{-m\nu/2}, \tag{56}$$

where the second-to-last line was obtained by noting that $j-1$ covers the same range of integers as $m-j-1$ under the product. Finally, using the definition of the multivariate Gamma function,

$$\prod_j \Gamma\left(\tfrac{\nu-j+1}{2}\right) \prod_{k \in \{j+1,\dots,m\}} \sqrt{\pi} = \pi^{m(m-1)/4} \prod_j \Gamma\left(\tfrac{\nu-j+1}{2}\right) = \Gamma_m\left(\tfrac{\nu}{2}\right). \tag{57}$$

We thereby re-obtain the probability density for the standard Wishart distribution,

$$\mathrm{P}\left(\boldsymbol{\Sigma}\right) = \frac{|\boldsymbol{\Sigma}|^{(\nu-m-1)/2} e^{-\operatorname{Tr}(\boldsymbol{\Sigma})/2}}{2^{m\nu/2}\Gamma_m\left(\tfrac{\nu}{2}\right)}. \tag{58}$$

## D    FULL DESCRIPTION OF OUR METHOD FOR DEEP GAUSSIAN PROCESS

Here we give the full derivation for our doubly-stochastic variational deep GPs, following Salimbeni & Deisenroth (2017). A deep Gaussian process (DGP; (Damianou & Lawrence, 2013; Salimbeni & Deisenroth, 2017)) defines a prior over function values, $\mathbf{F}_\ell \in \mathbb{R}^{P \times N_\ell}$, where $\ell$ is the layer, $P$ is the number of input points, and $N_\ell$ is the "width" of this layer, by stacking $L+1$ layers of standard Gaussian processes (we use $L+1$ layers instead of the typical $L$ layers to retain consistency with how we define BNNs):

$$\mathrm{P}\left(\mathbf{Y}, \{\mathbf{F}_\ell\}_{\ell=1}^{L+1} | \mathbf{X}\right) = \mathrm{P}\left(\mathbf{Y}|\mathbf{F}_L\right) \prod_{\ell=1}^{L+1} \mathrm{P}\left(\mathbf{F}_\ell|\mathbf{F}_{\ell-1}\right). \tag{59}$$

Here, the input is $\mathbf{F}_0 = \mathbf{X}$, and the output is $\mathbf{Y}$ (which could be continuous values for regression, or class labels for classification), and the distribution over $\mathbf{F}_\ell \in \mathbb{R}^{P \times N_\ell}$ factorises into independent multivariate Gaussian distributions over each function,

$$\mathrm{P}\left(\mathbf{F}_\ell|\mathbf{F}_{\ell-1}\right) = \prod_{\lambda=1}^{N_\ell} \mathrm{P}\left(\mathbf{f}_\lambda^\ell|\mathbf{F}_{\ell-1}\right) = \prod_{\lambda=1}^{N_\ell} \mathcal{N}\left(\mathbf{f}_\lambda^\ell|\mathbf{0}, \mathbf{K}\left(\mathbf{F}_{\ell-1}\right)\right), \tag{60}$$

where $\mathbf{f}_\lambda^\ell$ is the $\lambda$th column of $\mathbf{F}_\ell$, giving the activation of all datapoints for the $\lambda$th feature, and $\mathbf{K}\left(\cdot\right)$ is a function that computes the kernel-matrix from the features in the previous layer.

To define a variational approximate posterior, we augment $\{\mathbf{F}_\ell\}_{\ell=1}^{L+1}$ with inducing points consisting of the function values $\{\mathbf{U}_\ell\}_{\ell=1}^{L+1}$,

$$\mathrm{P}\left(\mathbf{Y}, \{\mathbf{F}_\ell, \mathbf{U}_\ell\}_{\ell=1}^{L+1} | \mathbf{X}, \mathbf{U}_0\right) = \mathrm{P}\left(\mathbf{Y}|\mathbf{F}_{L+1}\right) \prod_{\ell=1}^{L+1} \mathrm{P}\left(\mathbf{F}_\ell, \mathbf{U}_\ell|\mathbf{F}_{\ell-1}, \mathbf{U}_{\ell-1}\right) \tag{61}$$

and because $\mathbf{F}_\ell$ and $\mathbf{U}_\ell$ are the function outputs corresponding to different inputs ($\mathbf{F}_{\ell-1}$ and $\mathbf{U}_{\ell-1}$), they form a joint multivariate Gaussian distribution, analogous to Eq. (60),

$$\mathrm{P}\left(\mathbf{F}_\ell, \mathbf{U}_\ell|\mathbf{F}_{\ell-1}, \mathbf{U}_{\ell-1}\right) = \prod_{\lambda=1}^{N_l} \mathcal{N}\left(\begin{pmatrix} \mathbf{f}_\lambda^\ell \\ \mathbf{u}_\lambda^\ell \end{pmatrix}\bigg|\mathbf{0}, \mathbf{K}\left(\begin{pmatrix} \mathbf{F}_{\ell-1} \\ \mathbf{U}_{\ell-1} \end{pmatrix}\right)\right). \tag{62}$$

Following Salimbeni & Deisenroth (2017) we form an approximate posterior by conditioning the function values, $\mathbf{F}_\ell$ on the inducing outputs, $\mathbf{U}_\ell$,

$$\begin{aligned}
\mathrm{Q}\left(\{\mathbf{F}_\ell, \mathbf{U}_\ell\}_{\ell=1}^{L+1}\big|\mathbf{X}, \mathbf{U}_0\right) &= \textstyle\prod_{\ell=1}^{L+1} \mathrm{Q}\left(\mathbf{F}_\ell, \mathbf{U}_\ell|\mathbf{F}_{\ell-1}, \mathbf{U}_{\ell-1}\right) \\
&= \textstyle\prod_{\ell=1}^{L+1} \mathrm{P}\left(\mathbf{F}_\ell|\mathbf{U}_\ell, \mathbf{U}_{\ell-1}, \mathbf{F}_{\ell-1}\right) \mathrm{Q}\left(\mathbf{U}_\ell|\mathbf{U}_{\ell-1}\right),
\end{aligned} \tag{63}$$

where $\mathrm{Q}\left(\mathbf{U}_\ell|\mathbf{U}_{\ell-1}\right)$ is given by Eq. (19), i.e.

$$\mathrm{Q}\left(\mathbf{U}_\ell|\mathbf{U}_{\ell-1}\right) = \textstyle\prod_{\lambda=1}^{N_\ell} \mathcal{N}\left(\mathbf{u}_\lambda^\ell \big| \boldsymbol{\Sigma}_\ell^{\mathbf{u}} \boldsymbol{\Lambda}_\ell \mathbf{v}_\lambda^\ell, \boldsymbol{\Sigma}_\ell^{\mathbf{u}}\right), \qquad \boldsymbol{\Sigma}_\ell^{\mathbf{u}} = \left(\mathbf{K}^{-1}\left(\mathbf{U}_{\ell-1}\right) + \boldsymbol{\Lambda}_\ell\right)^{-1}, \tag{64}$$

and $\mathrm{P}\left(\mathbf{F}_\ell|\mathbf{U}_\ell, \mathbf{U}_{\ell-1}, \mathbf{F}_{\ell-1}\right)$ is given by standard manipulations of Eq. (62). Importantly, the prior can be factorised in an analogous fashion,

$$\mathrm{P}\left(\mathbf{F}_\ell, \mathbf{U}_\ell|\mathbf{F}_{\ell-1}, \mathbf{U}_{\ell-1}\right) = \mathrm{P}\left(\mathbf{F}_\ell|\mathbf{U}_\ell, \mathbf{U}_{\ell-1}, \mathbf{F}_{\ell-1}\right) \mathrm{P}\left(\mathbf{U}_\ell|\mathbf{U}_{\ell-1}\right). \tag{65}$$

The full ELBO can be written as

$$\mathcal{L} = \mathbb{E}_{\mathrm{Q}\left(\{\mathbf{F}_\ell, \mathbf{U}_\ell\}_{\ell=1}^{L+1}\big|\mathbf{X}, \mathbf{U}_0\right)} \left[\log \frac{\mathrm{P}\left(\mathbf{Y}, \{\mathbf{F}_\ell, \mathbf{U}_\ell\}_{\ell=1}^{L+1}\big|\mathbf{X}, \mathbf{U}_0\right)}{\mathrm{Q}\left(\{\mathbf{F}_\ell, \mathbf{U}_\ell\}_{\ell=1}^{L+1}\big|\mathbf{X}, \mathbf{U}_0\right)}\right]. \tag{66}$$

Substituting Eqs. (61), (63) and (65) into Eq. (66), the $\mathrm{P}\left(\mathbf{F}_\ell|\mathbf{U}_\ell, \mathbf{U}_{\ell-1}, \mathbf{F}_{\ell-1}\right)$ terms cancel and we obtain,

$$\mathcal{L} = \mathbb{E}_{\mathrm{Q}\left(\{\mathbf{F}_\ell, \mathbf{U}_\ell\}_{\ell=1}^{L+1}\big|\mathbf{X}, \mathbf{U}_0\right)} \left[\log \mathrm{P}\left(\mathbf{Y}|\mathbf{F}_{L+1}\right) + \sum_{\ell=1}^{L+1} \log \frac{\mathrm{P}\left(\mathbf{U}_\ell|\mathbf{U}_{\ell-1}\right)}{\mathrm{Q}\left(\mathbf{U}_\ell|\mathbf{U}_{\ell-1}\right)}\right]. \tag{67}$$

Analogous to the BNN case, we evaluate the ELBO by alternatively sampling the inducing function values $\mathbf{U}_\ell$ given $\mathbf{U}_{\ell-1}$ and propagating the data by sampling from $\mathrm{P}\left(\mathbf{F}_\ell|\mathbf{U}_\ell, \mathbf{U}_{\ell-1}, \mathbf{F}_{\ell-1}\right)$ (see Alg. 2). As in Salimbeni & Deisenroth (2017), since all likelihoods we use factorise across datapoints, we only need to sample from the marginals of the latter distribution to sample the propagated function values. As in the BNN case, the parameters of the approximate posterior are the global inducing inputs, $\mathbf{U}_0$, and the pseudo-data and precisions at all layers, $\{\mathbf{V}_\ell, \boldsymbol{\Lambda}_\ell\}_{\ell=1}^L$. We can similarly use standard reparameterised variational inference (Kingma & Welling, 2013; Rezende et al., 2014) to optimise these variational parameters, as $\mathrm{Q}\left(\mathbf{U}_\ell|\mathbf{U}_{\ell-1}\right)$ is Gaussian. We use Adam (Kingma & Ba, 2014) to optimise the parameters.

---

**Algorithm 2:** Global inducing points for deep Gaussian processes

---

**Parameters:** inducing inputs, $\mathbf{U}_0$, inducing outputs and precisions, $\{\mathbf{V}_\ell, \boldsymbol{\Lambda}_\ell\}_{\ell=1}^L$, at all layers.
**DGP inputs:** (e.g. MNIST digits) $\mathbf{F}_0$
**DGP outputs:** (e.g. classification logits) $\mathbf{F}_{L+1}$
$\mathcal{L} \leftarrow 0$
**for** $\ell \in \{1, \ldots, L+1\}$ **do**
    *Compute the mean and covariance over the inducing outputs at this layer*
    $\boldsymbol{\Sigma}_\ell^{\mathbf{u}} = \left(\mathbf{K}^{-1}\left(\mathbf{U}_{\ell-1}\right) + \boldsymbol{\Lambda}_\ell\right)^{-1}$
    $\mathbf{M}_\ell = \boldsymbol{\Sigma}_\ell^{\mathbf{u}} \boldsymbol{\Lambda}_\ell \mathbf{V}_\ell$
    *Sample the inducing outputs and compute the ELBO*
    $\mathbf{U}_\ell \sim \mathcal{N}\left(\mathbf{M}_\ell, \boldsymbol{\Sigma}_\ell^{\mathbf{u}}\right) = \mathrm{Q}\left(\mathbf{U}_\ell|\mathbf{U}_{\ell-1}\right)$
    $\mathcal{L} \leftarrow \mathcal{L} + \log \mathrm{P}\left(\mathbf{U}_\ell|\mathbf{U}_{\ell-1}\right) - \log \mathcal{N}\left(\mathbf{U}_\ell|\mathbf{M}_\ell, \boldsymbol{\Sigma}_\ell^{\mathbf{u}}\right)$
    *Propagate the inputs using the sampled inducing outputs,*
    $\mathbf{F}_\ell \sim \mathrm{P}\left(\mathbf{F}_\ell|\mathbf{U}_\ell, \mathbf{U}_{\ell-1}, \mathbf{F}_{\ell-1}\right)$
$\mathcal{L} \leftarrow \mathcal{L} + \log \mathrm{P}\left(\mathbf{Y}|\mathbf{F}_{L+1}\right)$

---

## E   Motivating the approximate posterior for deep GPs

Our original motivation for the approximate posterior was for the BNN case, which we then extended to deep GPs. Here, we show how the same approximate posterior can be motivated from a deep GP

perspective. As with the BNN case, we first derive the form of the optimal approximate posterior for the last layer, in the regression case. Without inducing points, the ELBO becomes

$$\mathcal{L} = \mathbb{E}_{Q\left(\{\mathbf{F}_\ell\}_{\ell=1}^{L+1}\right)}\left[\log \frac{P\left(\mathbf{Y}, \{\mathbf{F}_\ell\}_{\ell=1}^{L+1}\right)}{Q\left(\{\mathbf{F}_\ell\}_{\ell=1}^{L+1}\right)}\right], \quad (68)$$

where we have defined a generic variational posterior $Q\left(\{\mathbf{F}_\ell\}_{\ell=1}^{L+1}\right)$. Since we are interested in the form of $Q\left(\mathbf{F}_{L+1}\big|\{\mathbf{F}_\ell\}_{\ell=1}^L\right)$, we rearrange the ELBO so that all terms that do not depend on $\mathbf{F}_{L+1}$ are absorbed into a constant, $c$:

$$\mathcal{L} = \mathbb{E}_{Q\left(\{\mathbf{F}_\ell\}_{\ell=1}^{L+1}\right)}\left[\log P\left(\mathbf{Y}, \mathbf{F}_{L+1}\big|\{F_\ell\}_{\ell=1}^L\right) - \log Q\left(\mathbf{F}_{L+1}\right) + c\right]. \quad (69)$$

Some straightforward rearrangements lead to a similar form to before,

$$\mathcal{L} = \mathbb{E}_{Q\left(\{\mathbf{F}_\ell\}_{\ell=1}^L\right)}\big[\log P\left(\mathbf{Y}\big|\{\mathbf{F}_\ell\}_{\ell=1}^L\right)$$
$$- D_{KL}\left(Q\left(\mathbf{F}_{L+1}\big|\{\mathbf{F}_\ell\}_{\ell=1}^L\right) \big\| P\left(\mathbf{F}_{L+1}\big|\mathbf{Y}, \{\mathbf{F}_\ell\}_{\ell=1}^L\right)\right) + c\big], \quad (70)$$

from which we see that the optimal conditional posterior is given by $Q\left(\mathbf{F}_{L+1}\big|\{\mathbf{F}_\ell\}_{\ell=1}^L\right) = Q\left(\mathbf{F}_{L+1}\big|\mathbf{F}_L\right) = P\left(\mathbf{F}_{L+1}\big|\mathbf{Y}, \mathbf{F}_L\right)$, which has a closed form for regression: it is simply the standard GP posterior given by training data $\mathbf{Y}$ at inputs $\mathbf{F}_L$. In particular, for likelihood

$$P\left(\mathbf{Y}|\mathbf{F}_{L+1}, \mathbf{\Lambda}_{L+1}\right) = \prod_{\lambda=1}^{N_{L+1}} \mathcal{N}\left(\mathbf{y}_\lambda^{L+1}\big|\mathbf{f}_\lambda^{L+1}, \mathbf{\Lambda}_{L+1}^{-1}\right), \quad (71)$$

where $\mathbf{\Lambda}_{L+1}$ is the precision,

$$Q\left(\mathbf{F}_{L+1}|\mathbf{F}_L\right) = \prod_{\lambda=1}^{N_{L+1}} \mathcal{N}\left(\mathbf{f}_\lambda^{L+1}\big|\mathbf{\Sigma}_{L+1}^{\mathbf{f}}\mathbf{\Lambda}_{L+1}\mathbf{y}_\lambda^{L+1}, \mathbf{\Sigma}_{L+1}^{\mathbf{f}}\right), \quad (72)$$

$$\mathbf{\Sigma}_{L+1}^{\mathbf{f}} = \left(\mathbf{K}^{-1}\left(\mathbf{F}_L\right) + \mathbf{\Lambda}_{L+1}\right)^{-1}. \quad (73)$$

Finally, as is usual in GPs (Rasmussen & Williams, 2006), the predictive distribution for test points can be obtained by conditioning using this approximate posterior.

## F   COMPARING OUR DEEP GP APPROXIMATE POSTERIOR TO PREVIOUS WORK

The standard approach to inference in deep GPs (e.g. Salimbeni & Deisenroth, 2017) involves local inducing points and an approximate posterior over $\{\mathbf{U}_\ell\}_{\ell=1}^{L+1}$ that is factorised across layers,

$$Q\left(\{\mathbf{U}_\ell\}_{\ell=1}^{L+1}\right) = \prod_{l=1}^{L+1}\prod_{\lambda=1}^{N_\ell} \mathcal{N}\left(\mathbf{u}_\lambda^\ell; \mathbf{m}_\lambda^\ell, \mathbf{\Sigma}_\lambda^\ell\right). \quad (74)$$

This approximate posterior induces an approximate posterior over the underlying infinite-dimensional functions $\mathcal{F}_\ell$ at each layer, which are implicitly used to propagate the data through the network via $\mathbf{F}_\ell = \mathcal{F}_\ell(\mathbf{F}_{\ell-1})$. We show a graphical model summarising the standard approach in Fig. A1a. While, as Salimbeni & Deisenroth (2017) point out, the function values $\{\mathbf{F}_\ell\}_{\ell=1}^{L+1}$ are correlated, the functions $\{\mathcal{F}_\ell\}_{\ell=1}^{L+1}$ themselves are independent across layers. We note that for BNNs, this is equivalent to having a posterior over weights that factorises across layers.

One approach to introduce dependencies across layers for the functions would be to introduce the notion of global inducing points, propagating the initial $\mathbf{U}_0$ through the model. In fact, as we note in the Related Work section, Ustyuzhaninov et al. (2020) independently proposed this approach to introducing dependencies, using a toy problem to motivate the approach. However, they kept the form of the approximate posterior the same as the standard approach (Eq. 18); we show the corresponding graphical model in Fig. A1b. The graphical model shows that as adjacent functions $\mathcal{F}_\ell$ and $\mathcal{F}_{\ell+1}$ share the parent node $\mathbf{U}_\ell$, they are in fact dependent. However, non-adjacent functions do not share any parent nodes, and so are independent; this can be seen by considering the d-separation criterion (Pearl, 1988) for $\mathcal{F}_{\ell-1}$ and $\mathcal{F}_{\ell+1}$, which have parents $(\mathbf{U}_{\ell-2}, \mathbf{U}_{\ell-1})$ and $(\mathbf{U}_\ell, \mathbf{U}_{\ell+1})$ respectively.

Our approach, by contrast, determines the form of the approximate posterior over $\mathbf{U}_\ell$ by performing Bayesian regression using $\mathbf{U}_{\ell-1}$ as input to that layer's GP, where the output data is $\mathbf{V}_\ell$. This results in a posterior that depends on the previous layer, $Q\left(\mathbf{U}_\ell|\mathbf{U}_{\ell-1}\right)$. We show the corresponding graphical model in Fig. A1c. From this graphical model it is straightforward to see that our approach results in a posterior over functions that are correlated across all layers.

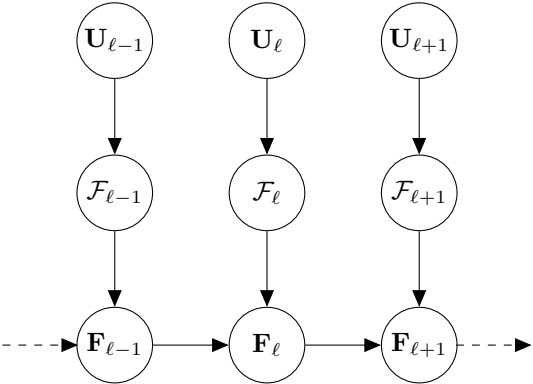

(a) Graphical model illustrating the approach of Salimbeni & Deisenroth (2017). The inducing inputs, $\{\mathbf{U}_{\ell-1}\}_{\ell=1}^{L+1}$ are treated as learned parameters and are therefore omitted from the model.

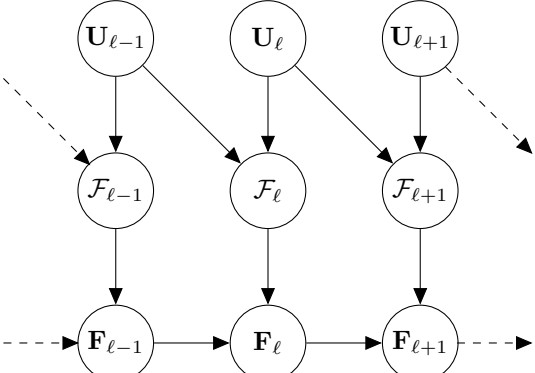

(b) Graphical model illustrating the approach of Ustyuzhaninov et al. (2020). The inducing inputs are given by the inducing outputs at the previous layer.

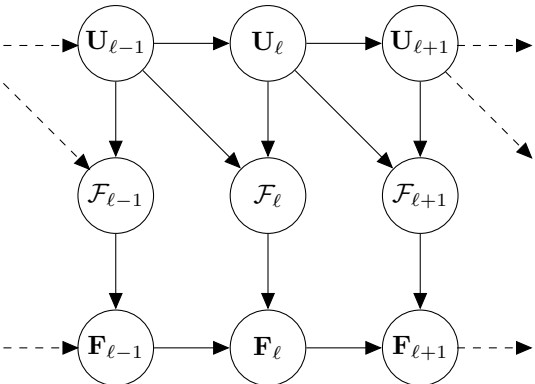

(c) Graphical model illustrating our approach.

Figure A1: Comparison of the graphical models for three approaches to inference in deep GPs: Salimbeni & Deisenroth (2017), Ustyuzhaninov et al. (2020), and ours.

## G    PARAMETER SCALING FOR ADAM

The standard optimiser for variational BNNs is ADAM (Kingma & Ba, 2014), which we also use. Considering similar RMSprop updates for simplicity (Tieleman & Hinton, 2012),

$$\Delta w = \eta \frac{g}{\sqrt{\mathbb{E}\left[g^2\right]}} \tag{75}$$

where the expectation over $g^2$ is approximated using a moving-average of past gradients. Thus, absolute parameter changes are going to be of order $\eta$. This is fine if all the parameters have roughly

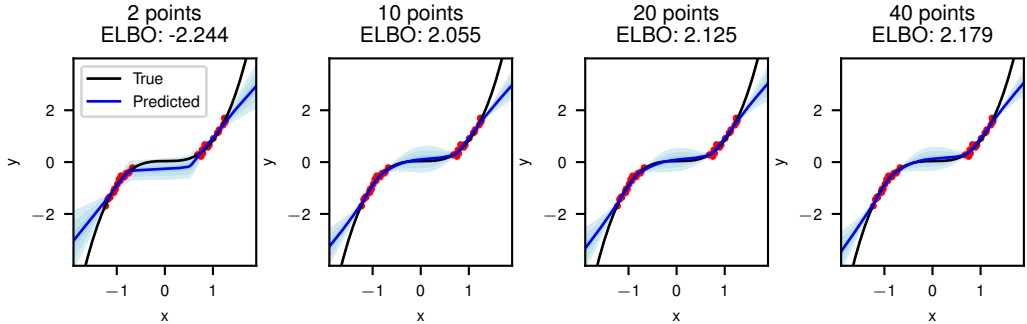

Figure A2: Predictive distributions on the toy dataset as the number of inducing points changes.

the same order of magnitude, but becomes a serious problem if some of the parameters are very large and others are very small. For instance, if a parameter is around $10^{-4}$ and $\eta = 10^{-4}$, then a single ADAM step can easily double the parameter estimate, or change it from positive to negative. In contrast, if a parameter is around $1$, then ADAM, with $\eta^{-4}$ can make proportionally much smaller changes to this parameter, (around $0.01\%$). Thus, we need to ensure that all of our parameters have the same scale, especially as we mix methods, such as combining factorised and global inducing points. We thus design all our new approximate posteriors (i.e. the inducing inputs and outputs) such that the parameters have a scale of around $1$. The key issue is that the mean weights in factorised methods tend to be quite small — they have scale around $1/\sqrt{\text{fan-in}}$. To resolve this issue, we store scaled weights, and we divide these stored, scaled mean parameters by the fan-in as part of the forward pass,

$$\text{weights} = \frac{\text{scaled weights}}{\sqrt{\text{fan-in}}}. \tag{76}$$

This scaling forces us to use larger learning rates than are typically used.

## H EXPLORING THE EFFECT OF THE NUMBER OF INDUCING POINTS

In this section, we briefly consider the effect of changing the number of inducing points, $M$, used in global inducing. We reconsider the toy problem from Sec. 4.1, and plot predictive posteriors obtained with global inducing as the number of inducing points increases from 2 to 40 (noting that in Fig. 1 we used 100 inducing points).

We plot the results of our experiment in Fig. A2. While two inducing points are clearly not sufficient, we observe that there is remarkably very little difference between the predictive posteriors for 10 or more inducing points. This observation is reflected in the ELBOs per datapoint (listed above each plot), which show that adding more points beyond 10 gains very little in terms of closeness to the true posterior.

However, we note that this is a very simple dataset: it consists of only two clusters of close points with a very clear trend. Therefore, we would expect that for more complex datasets more inducing points would be necessary. We leave a full investigation of how many inducing points are required to obtain a suitable approximate posterior, such as that found in Burt et al. (2020) for sparse GP regression, to future work.

## I UNDERSTANDING COMPOSITIONAL UNCERTAINTY

In this section, we take inspiration from the experiments of Ustyuzhaninov et al. (2020) which investigate the compositional uncertainty obtained by different approximate posteriors for DGPs. They noted that methods which factorise over layers have a tendency to cause the posterior distribution for each layer to collapse to a (nearly) deterministic function, resulting in worse uncertainty quantification within layers and worse ELBOs. In contrast, they found that allowing the approximate posterior to have correlations between layers allows those layers to capture more uncertainty, resulting in better

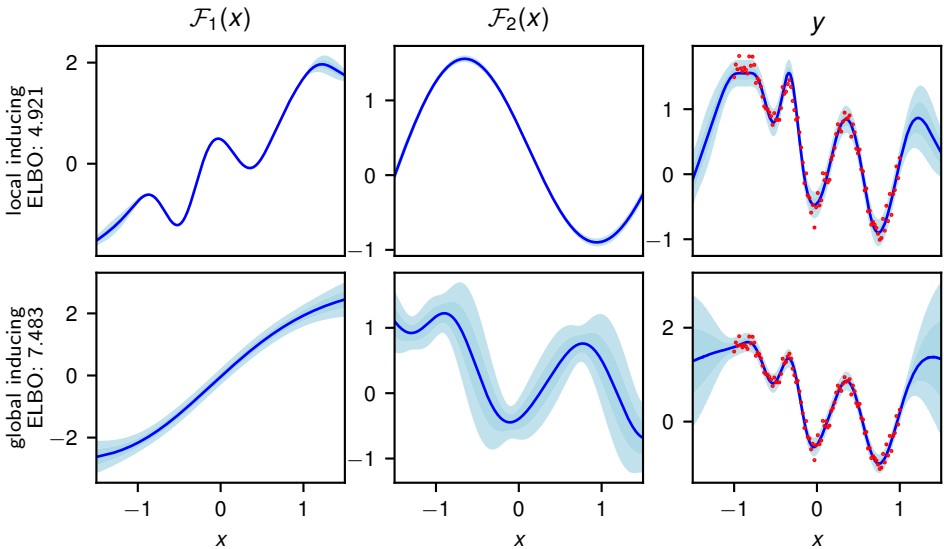

Figure A3: Posterior distributions for 2-layer DGPs with local inducing and global inducing. The first two columns show the predictive distributions for each layer taken individually, while the last column shows the predictive distribution of the output $y$.

Table 2: ELBOs and variances of the intermediate functions for a BNN fit to the toy data of Fig. 1.

|  | ELBO | $\mathbb{V}[\mathcal{F}_1]$ | $\mathbb{V}[\mathcal{F}_2]$ | $\mathbb{V}[\mathcal{F}_3]$ |
|---|---|---|---|---|
| factorised | -4.585 | 0.0728 | 0.4765 | 0.1926 |
| local inducing | -5.469 | 0.0763 | 0.4473 | 0.0643 |
| global inducing | 2.236 | 0.4820 | 0.4877 | 1.0820 |

ELBOs and therefore a closer approximation to the true posterior. They argue that this then allows the model to better discover compositional structure in the data.

We first consider a toy problem consisting of 100 datapoints generated by sampling from a two-layer DGP of width one, with squared-exponential kernels in each layer. We then fit two two-layer DGPs to this data - one using local inducing, the other using global inducing. The results of this experiment can be seen in Fig. A3, which show the final fit, along with the learned posteriors over intermediate functions $\mathcal{F}_1$ and $\mathcal{F}_2$. These results mirror those observed by Ustyuzhaninov et al. (2020) on a similar experiment: local inducing, which factorises over layers, collapses to a nearly deterministic posterior over the intermediate functions, whereas global inducing provides a much broader distribution over functions for the two layers. Therefore, global inducing leads to a wider range of plausible functions that could explain the data via composition, which can be important in understanding the data. We observe that this behaviour directly leads to better uncertainty quantification for the out-of-distribution region, as well as better ELBOs.

To illustrate a similar phenomenon in BNNs, we reconsider the toy problem of Sec. 4.1. As it is not meaningful to consider neural networks with only one hidden unit per layer, instead of plotting intermediate functions we instead look at the mean variance of the functions at random input points, following roughly the experiment Dutordoir et al. (2019) in Table 1. For each layer, we consider the quantity

$$\mathbb{E}_x \left[ \frac{1}{N_\ell} \sum_{\lambda=1}^{N_\ell} \mathbb{V}[f_\lambda^l(x)] \right], \tag{77}$$

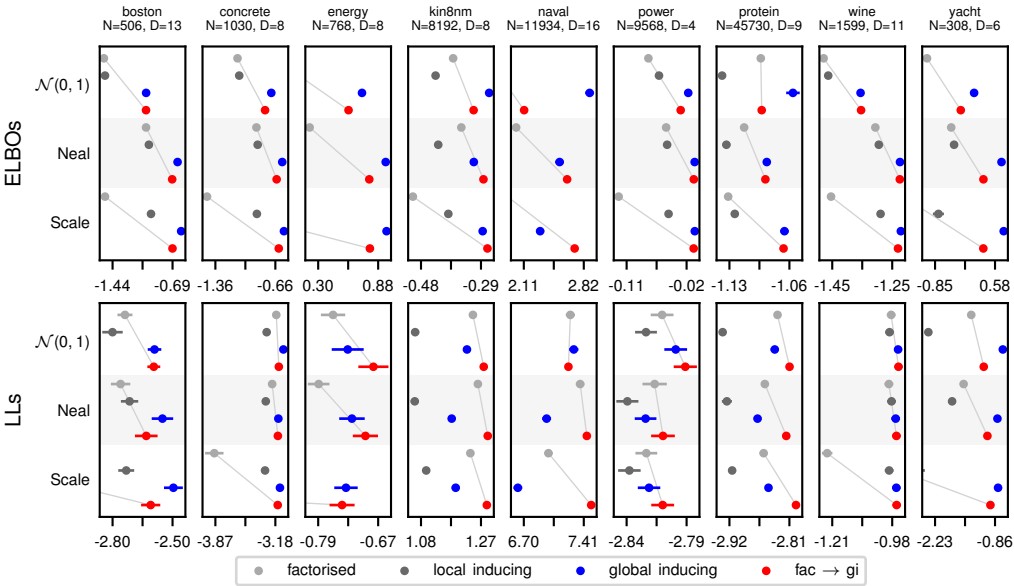

Figure A4: ELBOs per datapoint and average test log likelihoods for BNNs on UCI datasets.

where the expectation is over random input points, which we sample from a standard normal. We expect that for methods which introduce correlations across layers, this quantity will be higher, as there will be a wider range of intermediate functions that could plausibly explain the data. We confirm this in Table 2, which indicates that global inducing leads to many more compositions of functions being considered as plausible explanations of the data. This is additionally reflected in the ELBO, which is far better for global inducing than the other, factorised methods. However, we note that the variances are far closer than we might otherwise expect for the second layer. We hypothesise that this is due to the pruning effects described in Trippe & Turner (2018), where a layer has many weights that are close to the prior that are then pruned out by the following layer by having the outgoing weights collapse to zero. In fact, we note that the variances in the last layer are small for the factorised methods, which supports this hypothesis. By contrast, global inducing leads to high variances across all layers.

We believe that understanding the role of compositional uncertainty in variational inference for deep Bayesian models can lead to important conclusions about both the models being used and the compositional structure underlying the data being modelled, and is therefore an important direction for future work to consider.

## J    UCI RESULTS WITH BAYESIAN NEURAL NETWORKS

For this Appendix, we consider all of the UCI datasets from Hernández-Lobato & Adams (2015), along with four approximation families: factorised (i.e. mean-field), local inducing, global inducing, and fac→gi, which may offer some computational advantages to global inducing. We also considered three priors: the standard $\mathcal{N}(0, 1)$ prior, NealPrior, and ScalePrior. The test LLs and ELBOs for BNNs applied to UCI datasets are given in Fig. A4. Note that the ELBOs for the global inducing methods (both global inducing and fac→gi) are almost always better than those for baseline methods, often by a very large margin. However, as noted earlier, this does not necessarily correspond to better test log likelihoods due to model misspecification: there is not a straightforward relationship between the ELBO and the predictive performance, and so it is possible to obtain better test log likelihoods with worse inference. We present all the results, including for the test RMSEs, in tabulated form in Appendix P.

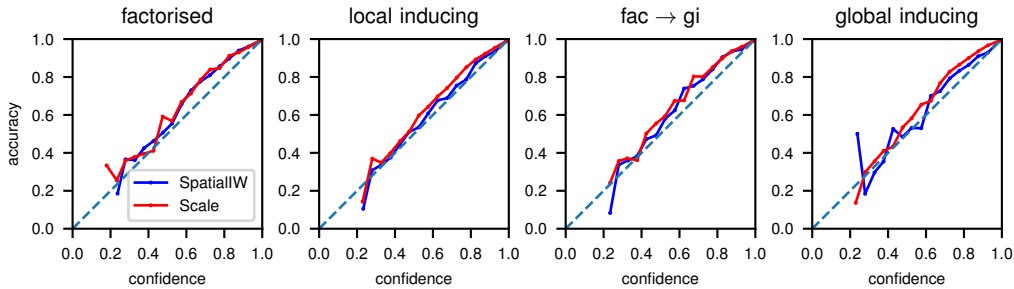

Figure A5: Calibration curves for CIFAR-10

## J.1 EXPERIMENTAL DETAILS

The architecture we considered for all BNN UCI experiments were fully-connected ReLU networks with 2 hidden layers of 50 hidden units each. We performed a grid search to select the learning rate and minibatch size. For the fully factorised approximation, we selected the learning rate from {3e-4, 1e-3, 3e-3, 1e-2} and the minibatch size from {32, 100, 500}, optimising for 25000 gradient steps; for the other methods we selected the learning rate from {3e-3, 1e-2} and fixed the minibatch size to 10000 (as in Salimbeni & Deisenroth (2017)), optimising for 10000 gradient steps. For all methods we selected the hyperparameters that gave the best ELBO. We trained the models using 10 samples from the approximate posterior, while using 100 for evaluation. For the inducing point methods, we used the selected batch size for the number of inducing points per layer. For all methods, we initialised the log noise variance at -3, but use the scaling trick in Appendix G to accelerate convergence, scaling by a factor of 10. Note that for the fully factorised method we used the local reparameterisation trick (Kingma et al., 2015); however, for fac → gi we cannot do so because the inducing point methods require that covariances be propagated through the network correctly. For the inducing point methods, we additionally use output channel-specific precisions, $\Lambda_{l,\lambda}$, which effectively allows the network to prune unnecessary neurons if that benefits the ELBO. However, we only parameterise the diagonal of these precision matrices to save on computational and memory cost.

## K  UNCERTAINTY CALIBRATION FOR CIFAR-10

To assess how well our methods capture uncertainty, we consider calibration. Calibration is assessed by comparing the model's probabilistic assessment of its confidence with its accuracy — the proportion of the time that it is actually correct. For instance, gathering model predictions with some confidence (e.g. softmax probabilities in the range 0.9 to 0.95), and looking at the accuracy of these predictions, we would expect the model to be correct with probability 0.925; a higher or lower value would represent miscalibration.

We begin by plotting calibration curves in Fig. A5, obtained by binning the predictions in 20 equal bins and assessing the mean accuracy of the binned predictions. For well-calibrated models, we expect the line to lie on the diagonal. A line above the diagonal indicates the model is underconfident (the model is performing better than it expects), whereas a line below the diagonal indicates it is overconfident (it is performing worse than it expects). While it is difficult to draw strong conclusions from these plots, it appears generally that factorised is poorly calibrated for both priors, that SpatialIWPrior generally improves calibration over ScalePrior, and that local inducing with SpatialIWPrior performs very well.

To come to more quantitative conclusions, we use expected calibration error (ECE; (Naeini et al., 2015; Guo et al., 2017)), which measures the expected absolute difference between the model's confidence and accuracy. Confirming the results from the plots (Fig. A5), we find that using the more sophisticated SpatialIWPrior gave considerable improvements in calibration. While, as expected, we find that our most accurate prior, SpatialIWPrior, in combination with global inducing points did very well (ECE of 0.021), the model with the best ECE is actually local inducing with SpatialIWPrior, albeit by a very small margin. We leave investigation of exactly why this is to future work. Finally, note our final ECE value of 0.021 is a considerable improvement over those for uncalibrated models

Table 3: Expected calibration error for CIFAR-10

|  | factorised | local inducing | fac $\rightarrow$ gi | global inducing |
|---|---|---|---|---|
| ScalePrior | 0.053 | 0.040 | 0.049 | **0.038** |
| SpatialIWPrior | 0.045 | **0.018** | 0.036 | 0.021 |

in Guo et al. (2017) (Table 1), which are in the region of 0.03-0.045 (although considerably better calibration can be achieved by post-hoc scaling of the model's confidence).

## L ASYMPTOTIC COMPLEXITY

In the deep GP case, the complexity for global inducing is exactly that of standard inducing point Gaussian processes, i.e. $\mathcal{O}(M^3 + PM^2)$ where $M$ is the number of inducing points, and $P$ can be taken to be the number of training inputs, or the size of a minibatch, as appropriate. The first term, $M^3$, comes from computing and sampling the posterior over $\mathbf{U}_\ell$ based on the inducing points (e.g. inverting the covariance). The second term, and $PM^2$ comes from computing the implied distribution over $\mathbf{F}_\ell$.

In the fully-connected BNN case, we have three terms, $\mathcal{O}(N^3 + MN^2 + PN^2)$. The first term, $N^3$, where $N$ corresponds to the width of the network, arises from taking the inverse of the covariance matrix in Eq. (9), but is also the complexity e.g. for propagating the inducing points from layer to the next (Eq. 8). The second term, $MN^2$, comes from computing that covariance in Eq. (9), by taking the product of input features with themselves. The third term $PN^2$ comes from multiplying the training inputs/minibatch by the sampled inputs (Eq. 1).

## M UCI RESULTS WITH DEEP GAUSSIAN PROCESSES

In this appendix, we again consider all of the UCI datasets from Hernández-Lobato & Adams (2015) for DGPs with depths ranging from two to five layers. We compare DSVI (Salimbeni & Deisenroth, 2017), local inducing, and global inducing. While local inducing uses the same inducing-point architecture as Salimbeni & Deisenroth (2017), the actual implementation and parameterisation is very different and in addition they used a complex architecture involving skip connections that we may not have matched exactly. As such, we do expect to see differences between local inducing and Salimbeni & Deisenroth (2017).

We show our results in Fig. A6. Here, the results are not as clear-cut as in the BNN case. For the smaller datasets (i.e. boston, concrete, energy, wine, yacht), global inducing generally outperforms both local inducing and DSVI, as noted in the main text, especially when considering the ELBOs. We do however observe that for power, protein, and one model for kin8nm, the local approaches sometimes outperform global inducing, even for the ELBOs. We believe this is due to the fact that relatively few inducing points were used (100), in combination with the fact that global inducing has far fewer variational parameters than the local approaches. This may make optimisation harder in the global inducing case, especially for larger datasets where the model uncertainty does not matter as much, as the posterior concentration will be stronger. Importantly, however, our results on CIFAR-10 indicate that these issues do not arise in very large-scale, high-dimensional datasets, which are of most interest for future work.

We provide tabulated results, including for RMSEs, in Appendix P.

### M.1 EXPERIMENTAL DETAILS

Here, we matched the experimental setup in Salimbeni & Deisenroth (2017) as closely as possible. In particular, we used 100 inducing points, and full-covariance observation noise. However, our parameterisation is still somewhat different from theirs, in part because our approximate posterior is defined in terms of noisy function-values, while their approximate posterior was defined in terms of the function-values themselves.

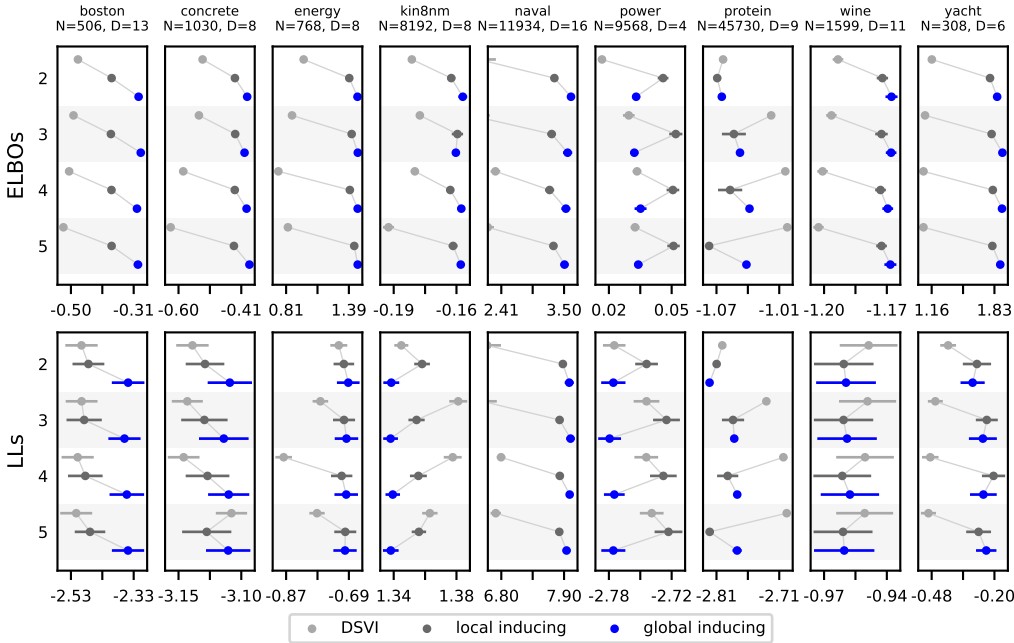

Figure A6: ELBOs per datapoint and average test log likelihoods for DGPs on UCI datasets. The numbers indicate the depths of the models.

As the original results in Salimbeni & Deisenroth (2017) used differ-ent UCI splits, and did not provide the ELBO, we reran their code `https://github.com/ICL-SML/Doubly-Stochastic-DGP` (changing the number of epochs and noise variance to reflect the values in the paper), which gave very similar log likelihoods to those in the paper.

## N  MNIST 500

For MNIST, we considered a LeNet-inspired model consisting of two conv2d-relu-maxpool blocks, followed by conv2d-relu-linear, where the convolutions all have $3 \times 3$ kernels with 64 channels. We trained all models using a learning rate of $10^{-3}$.

When training on very small datasets, such as the first 500 training examples in MNIST, we can see a variety of pathologies emerge with standard methods. To help build intuition for these pathologies, we introduce a sanity check for the ELBO. In particular, we could imagine a model that sets the distribution over all lower-layer parameters equal to the prior, and sets the top-layer parameters so as to ensure that the predictions are uniform. With 10 classes, this results in an average test log likelihood of $-2.30 \approx \log(1/10)$, and an ELBO (per datapoint) of approximately $-2.30$. We found that many combinations of the approximate posterior/prior converged to ELBOs near this baseline. Indeed, the only approximate posterior to escape this baseline for ScalePrior and SpatialIWPrior is global inducing points. This is because ScalePrior and SpatialIWPrior both offer the flexibility to shrink the prior variance, and hence shrink the weights towards zero, giving uniform predictions, and potentially zero KL divergence. In contrast, NealPrior and StandardPrior do not offer this flexibility: you always have to pay something in KL divergence in order to give uniform predictions. We believe that this is the reason that factorised performs better than expected with NealPrior, despite having an ELBO that is close to the baseline. Furthermore, it is unclear why local inducing gives very test log likelihood and performance, despite having an ELBO that is similar to factorised. For StandardPrior, all the ELBOs are far lower than the baseline, and far lower than for any other priors. Despite this, factorised and fac → gi in combination with StandardPrior appear to transiently perform better in terms of predictive accuracy than any other method. These results should sound a note of caution

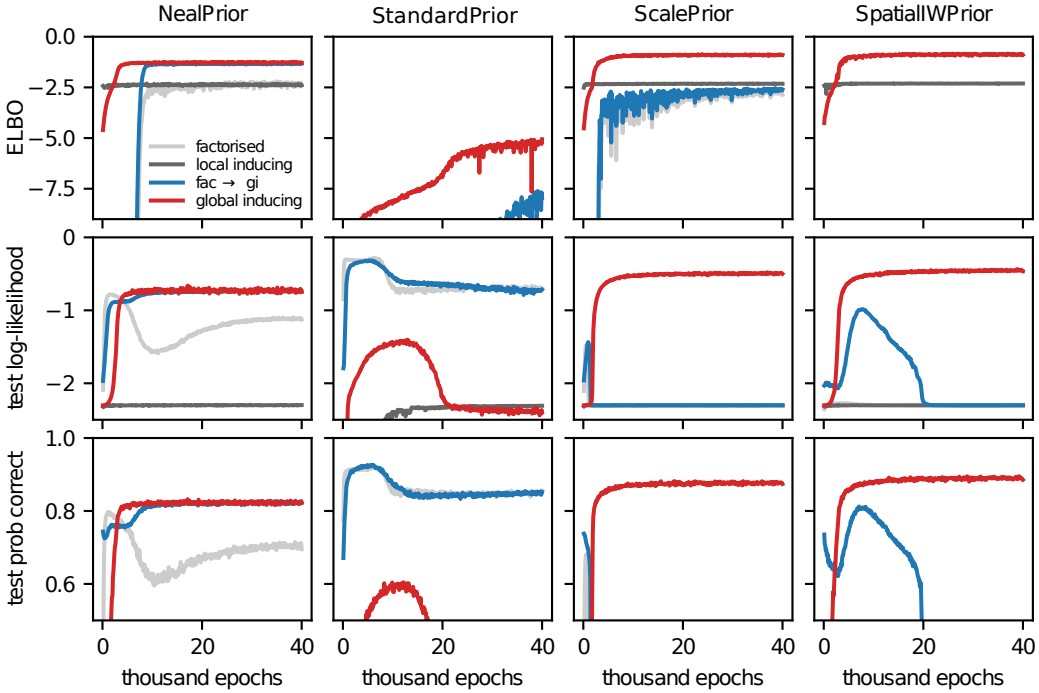

Figure A7: The ELBO, test log likelihoods and classification accuracy with different priors and approximate posteriors on a reduced MNIST dataset consisting of only the first 500 training examples.

whenever we try to use factorised approximate posteriors with fixed prior covariances (e.g. Blundell et al., 2015; Farquhar et al., 2020). We leave a full investigation of these effects for future work.

## O  ADDITIONAL EXPERIMENTAL DETAILS

All the methods were implemented in PyTorch. We ran the toy experiments on CPU, with the UCI experiments being run on a mixture of CPU and GPU. The remaining experiments – linear, CIFAR-10, and MNIST 500 – were run on various GPUs. For CIFAR-10, the most intensive of our experiments, we trained the models on one NVIDIA Tesla P100-PCIE-16GB. We optimised using ADAM (Kingma & Ba, 2014) throughout.

**Factorised**  We initialise the posterior weight means to follow the Neal scaling (i.e. drawn from NealPrior); however, we use the scaling described in Appendix G to accelerate training. We initialise the weight variances to $1\text{e-}3/\sqrt{N_{\ell-1}}$ for each layer.

**Inducing point methods**  For global inducing, we initialise the inducing inputs, $\mathbf{U}_0$, and pseudo-data for the last layer, $V_{L+1}$, using the first batch of data, except for the toy experiment, where we initialise using samples from $\mathcal{N}(0,1)$ (since we used more inducing points than datapoints). For the remaining layers, we initialise the pseudo-data by sampling from $\mathcal{N}(0,1)$. We initialise the log precision to $-4$, except for the last layer, where we initialise it to 0. We additionally use a scaling factor of 3 as described in Appendix G. For local inducing, the initialisation is largely the same, except we initialise the pseudo-data for every layer by sampling from $\mathcal{N}(0,1)$. We additionally sample the inducing inputs for every layer from $\mathcal{N}(0,1)$.

**Toy experiment**  For each variational method, we optimise the ELBO over 5000 epochs, using full batches for the gradient descent. We use a learning rate of 1e-2. We fix the noise variance at its

true value, to help assess the differences between each method more clearly. We use 10 samples from the variational posterior for training, using 100 for testing. For HMC, we use 10000 samples to burn in, and 10000 samples for evaluation, which we subsequently thin by a factor of 10. We initialise the samples from a standard normal distribution, and use 20 leapfrog steps for each sample. We hand-tune the leapfrog step sizes to be 0.0007 and 0.003 for the burn-in and sampling phases, respectively.

**Deep linear network** We use 10 inducing points for the inducing point methods. We use 1 sample from the approximate posterior for training and 10 for testing, training for 40 periods of 1000 gradient steps, using full batches for each step, with a learning rate of 1e-2.

**UCI experiments** The splits that we used for the UCI datasets can be found at `https://github.com/yaringal/DropoutUncertaintyExps`.

**CIFAR-10** The CIFAR-10 dataset (`https://www.cs.toronto.edu/~kriz/cifar.html`; (Krizhevsky et al., 2009)) is a 10-class dataset comprising RGB, $32 \times 32$ images. It is divided in two sets: a training set of 50,000 examples, and a validation set of 10,000 examples. For the purposes of this paper, we use the validation set as our test set and refer to it as such, as is commonly done in the literature. We use a batch size of 500, with one sample from the approximate posterior for training and 10 for testing. For pre-processing, we normalise the data using the training dataset's mean and standard deviation. Finally, we train for 1000 epochs with a learning rate of $1e - 2$ (see App. G for an explanation of why our learning rate is higher than might be expected), and we use a tempering scheme for the first 100 epochs, slowly increasing the influence of the KL divergence to the prior by multiplying it by a factor that increases from 0 to 1. In our scheme, we increase the factor in a step-wise manner, meaning that for the first ten epochs it is 0, then 0.1 for the next ten, 0.2 for the following ten, and so on. Importantly, we still have 900 epochs of training where the standard, untempered ELBO is used, meaning that our results reflect that ELBO. Finally, we note that we share the precisions $\Lambda_\ell$ within layers instead of using a separate precision for each output channel as was done in the UCI case. This saves memory and computational cost although possibly at the expense of predictive performance.

**MNIST 500** The MNIST dataset (`http://yann.lecun.com/exdb/mnist/`) is a dataset of grayscale handwritten digits, each $28 \times 28$ pixels, with 10 classes. It comprises 60,000 training images and 10,000 test images. For the MNIST 500 experiments, we trained using the first 500 images from the training dataset and discarded the rest. We normalised the images using the full training dataset's statistics.

**Deep GPs** As mentioned, we largely follow the approach of Salimbeni & Deisenroth (2017) for hyperparameters. For global inducing, we initialise the inducing inputs to the first batch of training inputs, and we initialise the pseudo-outputs for the last layer to the respective training outputs. For the remaining layers, we initialise the pseudo-outputs by sampling from a standard normal distribution. We initialise the precision matrix to be diagonal with log precision zero for the output layer, and log precision -4 for the remaining layers. For local inducing, we initialise inducing inputs and pseudo data by sampling from a standard normal for every layer, and initialise the precision matrices to be diagonal with log precision zero.

# P    TABLES OF UCI RESULTS

Table 4: Average test log likelihoods in nats for BNNs on UCI datasets (errors are $\pm$ 1 standard error)

|  | factorised | local inducing | global inducing | factorised → global |
|---|---|---|---|---|
| boston - $\mathcal{N}(0,1)$ | -2.74 ± 0.03 | -2.80 ± 0.04 | **−2.59 ± 0.03** | -2.60 ± 0.02 |
| NealPrior | -2.76 ± 0.04 | -2.71 ± 0.04 | **−2.55 ± 0.05** | -2.63 ± 0.05 |
| ScalePrior | -3.63 ± 0.03 | -2.73 ± 0.03 | **−2.50 ± 0.04** | -2.61 ± 0.04 |
| concrete - $\mathcal{N}(0,1)$ | -3.17 ± 0.02 | -3.28 ± 0.01 | **−3.08 ± 0.01** | -3.14 ± 0.01 |
| NealPrior | -3.21 ± 0.01 | -3.29 ± 0.01 | **−3.14 ± 0.01** | -3.15 ± 0.02 |
| ScalePrior | -3.89 ± 0.09 | -3.30 ± 0.01 | **−3.12 ± 0.01** | -3.15 ± 0.01 |
| energy - $\mathcal{N}(0,1)$ | -0.76 ± 0.02 | -1.75 ± 0.01 | -0.73 ± 0.03 | **−0.68 ± 0.03** |
| NealPrior | -0.79 ± 0.02 | -2.06 ± 0.09 | -0.72 ± 0.02 | **−0.70 ± 0.02** |
| ScalePrior | -2.55 ± 0.01 | -2.42 ± 0.02 | **−0.73 ± 0.02** | -0.74 ± 0.02 |
| kin8nm - $\mathcal{N}(0,1)$ | 1.24 ± 0.01 | 1.06 ± 0.01 | 1.22 ± 0.01 | **1.28 ± 0.01** |
| NealPrior | 1.26 ± 0.01 | 1.06 ± 0.01 | 1.18 ± 0.01 | **1.29 ± 0.01** |
| ScalePrior | 1.23 ± 0.01 | 1.10 ± 0.01 | 1.19 ± 0.01 | **1.29 ± 0.00** |
| naval - $\mathcal{N}(0,1)$ | 7.25 ± 0.04 | 6.06 ± 0.10 | **7.29 ± 0.04** | 7.23 ± 0.02 |
| NealPrior | 7.37 ± 0.03 | 4.28 ± 0.37 | 6.97 ± 0.04 | **7.45 ± 0.02** |
| ScalePrior | 6.99 ± 0.03 | 2.80 ± 0.00 | 6.63 ± 0.04 | **7.50 ± 0.02** |
| power - $\mathcal{N}(0,1)$ | -2.81 ± 0.01 | -2.82 ± 0.01 | -2.80 ± 0.01 | **−2.79 ± 0.01** |
| NealPrior | **−2.81 ± 0.01** | -2.84 ± 0.01 | -2.82 ± 0.01 | **−2.81 ± 0.01** |
| ScalePrior | -2.82 ± 0.01 | -2.84 ± 0.01 | -2.82 ± 0.01 | **−2.81 ± 0.01** |
| protein - $\mathcal{N}(0,1)$ | -2.83 ± 0.00 | -2.93 ± 0.00 | -2.84 ± 0.00 | **−2.81 ± 0.00** |
| NealPrior | -2.86 ± 0.00 | -2.92 ± 0.01 | -2.87 ± 0.00 | **−2.82 ± 0.00** |
| ScalePrior | -2.86 ± 0.00 | -2.91 ± 0.00 | -2.85 ± 0.00 | **−2.80 ± 0.00** |
| wine - $\mathcal{N}(0,1)$ | -0.98 ± 0.01 | -0.99 ± 0.01 | **−0.96 ± 0.01** | **−0.96 ± 0.01** |
| NealPrior | -0.99 ± 0.01 | -0.98 ± 0.01 | -0.97 ± 0.01 | **−0.96 ± 0.01** |
| ScalePrior | -1.22 ± 0.01 | -0.99 ± 0.01 | **−0.96 ± 0.01** | **−0.96 ± 0.01** |
| yacht - $\mathcal{N}(0,1)$ | -1.41 ± 0.05 | -2.39 ± 0.05 | **−0.68 ± 0.03** | -1.12 ± 0.02 |
| NealPrior | -1.58 ± 0.04 | -1.84 ± 0.05 | **−0.81 ± 0.03** | -1.04 ± 0.01 |
| ScalePrior | -4.12 ± 0.03 | -2.71 ± 0.22 | **−0.79 ± 0.02** | -0.97 ± 0.05 |

Table 5: Test RMSEs for BNNs on UCI datasets (errors are $\pm$ 1 standard error)

|  | factorised | local inducing | global inducing | factorised $\to$ global |
|---|---|---|---|---|
| boston - $\mathcal{N}(0,1)$ | $3.60 \pm 0.21$ | $3.85 \pm 0.26$ | $\mathbf{3.13 \pm 0.20}$ | $3.14 \pm 0.20$ |
| NealPrior | $3.64 \pm 0.24$ | $3.55 \pm 0.23$ | $\mathbf{3.14 \pm 0.18}$ | $3.33 \pm 0.21$ |
| ScalePrior | $9.03 \pm 0.26$ | $3.57 \pm 0.20$ | $\mathbf{2.97 \pm 0.19}$ | $3.27 \pm 0.20$ |
| concrete - $\mathcal{N}(0,1)$ | $5.73 \pm 0.11$ | $6.34 \pm 0.11$ | $\mathbf{5.39 \pm 0.09}$ | $5.55 \pm 0.10$ |
| NealPrior | $5.96 \pm 0.11$ | $6.35 \pm 0.12$ | $\mathbf{5.64 \pm 0.10}$ | $5.70 \pm 0.11$ |
| ScalePrior | $12.66 \pm 1.04$ | $6.48 \pm 0.12$ | $\mathbf{5.56 \pm 0.10}$ | $5.68 \pm 0.09$ |
| energy - $\mathcal{N}(0,1)$ | $0.51 \pm 0.01$ | $1.35 \pm 0.02$ | $0.50 \pm 0.01$ | $\mathbf{0.47 \pm 0.02}$ |
| NealPrior | $0.51 \pm 0.01$ | $1.95 \pm 0.14$ | $0.49 \pm 0.01$ | $\mathbf{0.47 \pm 0.01}$ |
| ScalePrior | $3.02 \pm 0.05$ | $2.67 \pm 0.06$ | $0.50 \pm 0.01$ | $\mathbf{0.49 \pm 0.01}$ |
| kin8nm - $\mathcal{N}(0,1)$ | $\mathbf{0.07 \pm 0.00}$ | $0.08 \pm 0.00$ | $\mathbf{0.07 \pm 0.00}$ | $\mathbf{0.07 \pm 0.00}$ |
| NealPrior | $\mathbf{0.07 \pm 0.00}$ | $0.08 \pm 0.00$ | $\mathbf{0.07 \pm 0.00}$ | $\mathbf{0.07 \pm 0.00}$ |
| ScalePrior | $\mathbf{0.07 \pm 0.00}$ | $0.08 \pm 0.00$ | $\mathbf{0.07 \pm 0.00}$ | $\mathbf{0.07 \pm 0.00}$ |
| naval - $\mathcal{N}(0,1)$ | $\mathbf{0.00 \pm 0.00}$ | $\mathbf{0.00 \pm 0.00}$ | $\mathbf{0.00 \pm 0.00}$ | $\mathbf{0.00 \pm 0.00}$ |
| NealPrior | $\mathbf{0.00 \pm 0.00}$ | $0.01 \pm 0.00$ | $\mathbf{0.00 \pm 0.00}$ | $\mathbf{0.00 \pm 0.00}$ |
| ScalePrior | $\mathbf{0.00 \pm 0.00}$ | $0.01 \pm 0.00$ | $\mathbf{0.00 \pm 0.00}$ | $\mathbf{0.00 \pm 0.00}$ |
| power - $\mathcal{N}(0,1)$ | $4.00 \pm 0.03$ | $4.06 \pm 0.03$ | $3.96 \pm 0.04$ | $\mathbf{3.93 \pm 0.04}$ |
| NealPrior | $4.03 \pm 0.04$ | $4.13 \pm 0.03$ | $4.06 \pm 0.03$ | $\mathbf{4.00 \pm 0.04}$ |
| ScalePrior | $4.06 \pm 0.04$ | $4.12 \pm 0.04$ | $4.05 \pm 0.04$ | $\mathbf{4.01 \pm 0.04}$ |
| protein - $\mathcal{N}(0,1)$ | $4.12 \pm 0.02$ | $4.54 \pm 0.02$ | $4.14 \pm 0.02$ | $\mathbf{4.04 \pm 0.01}$ |
| NealPrior | $4.21 \pm 0.01$ | $4.50 \pm 0.03$ | $4.27 \pm 0.02$ | $\mathbf{4.06 \pm 0.01}$ |
| ScalePrior | $4.22 \pm 0.02$ | $4.46 \pm 0.01$ | $4.19 \pm 0.02$ | $\mathbf{4.00 \pm 0.02}$ |
| wine - $\mathcal{N}(0,1)$ | $0.65 \pm 0.01$ | $0.65 \pm 0.01$ | $\mathbf{0.63 \pm 0.01}$ | $\mathbf{0.63 \pm 0.01}$ |
| NealPrior | $0.66 \pm 0.01$ | $0.65 \pm 0.01$ | $\mathbf{0.64 \pm 0.01}$ | $\mathbf{0.64 \pm 0.01}$ |
| ScalePrior | $0.82 \pm 0.01$ | $0.65 \pm 0.01$ | $\mathbf{0.64 \pm 0.01}$ | $\mathbf{0.64 \pm 0.01}$ |
| yacht - $\mathcal{N}(0,1)$ | $0.98 \pm 0.07$ | $2.35 \pm 0.13$ | $0.56 \pm 0.04$ | $\mathbf{0.50 \pm 0.04}$ |
| NealPrior | $1.15 \pm 0.07$ | $1.37 \pm 0.11$ | $\mathbf{0.57 \pm 0.04}$ | $0.63 \pm 0.05$ |
| ScalePrior | $14.55 \pm 0.59$ | $5.75 \pm 1.34$ | $\mathbf{0.56 \pm 0.04}$ | $0.63 \pm 0.04$ |

Table 6: ELBOs per datapoint in nats for BNNs on UCI datasets (errors are $\pm$ 1 standard error)

| | factorised | local inducing | global inducing | factorised $\to$ global |
|---|---|---|---|---|
| boston - $\mathcal{N}(0,1)$ | -1.55 $\pm$ 0.00 | -1.54 $\pm$ 0.00 | $-\mathbf{1.02 \pm 0.01}$ | -1.03 $\pm$ 0.00 |
| NealPrior | -1.03 $\pm$ 0.00 | -0.99 $\pm$ 0.00 | $-\mathbf{0.63 \pm 0.00}$ | -0.70 $\pm$ 0.00 |
| ScalePrior | -1.54 $\pm$ 0.00 | -0.96 $\pm$ 0.00 | $-\mathbf{0.59 \pm 0.00}$ | -0.70 $\pm$ 0.00 |
| concrete - $\mathcal{N}(0,1)$ | -1.10 $\pm$ 0.00 | -1.08 $\pm$ 0.00 | $-\mathbf{0.71 \pm 0.00}$ | -0.78 $\pm$ 0.00 |
| NealPrior | -0.88 $\pm$ 0.00 | -0.87 $\pm$ 0.00 | $-\mathbf{0.59 \pm 0.00}$ | -0.65 $\pm$ 0.00 |
| ScalePrior | -1.45 $\pm$ 0.01 | -0.88 $\pm$ 0.00 | $-\mathbf{0.57 \pm 0.00}$ | -0.63 $\pm$ 0.00 |
| energy - $\mathcal{N}(0,1)$ | -0.13 $\pm$ 0.02 | -0.53 $\pm$ 0.01 | $\mathbf{0.72 \pm 0.00}$ | 0.59 $\pm$ 0.00 |
| NealPrior | 0.21 $\pm$ 0.00 | -0.33 $\pm$ 0.04 | $\mathbf{0.95 \pm 0.00}$ | 0.79 $\pm$ 0.01 |
| ScalePrior | -1.12 $\pm$ 0.00 | -0.47 $\pm$ 0.01 | $\mathbf{0.96 \pm 0.01}$ | 0.80 $\pm$ 0.01 |
| kin8nm - $\mathcal{N}(0,1)$ | -0.38 $\pm$ 0.00 | -0.43 $\pm$ 0.00 | $-\mathbf{0.26 \pm 0.00}$ | -0.31 $\pm$ 0.00 |
| NealPrior | -0.35 $\pm$ 0.00 | -0.43 $\pm$ 0.00 | -0.31 $\pm$ 0.00 | $-\mathbf{0.28 \pm 0.00}$ |
| ScalePrior | -0.51 $\pm$ 0.00 | -0.39 $\pm$ 0.01 | -0.29 $\pm$ 0.00 | $-\mathbf{0.27 \pm 0.00}$ |
| naval - $\mathcal{N}(0,1)$ | 1.68 $\pm$ 0.01 | 1.65 $\pm$ 0.09 | $\mathbf{2.89 \pm 0.04}$ | 2.11 $\pm$ 0.02 |
| NealPrior | 2.02 $\pm$ 0.02 | -0.09 $\pm$ 0.34 | 2.53 $\pm$ 0.04 | $\mathbf{2.62 \pm 0.02}$ |
| ScalePrior | 1.91 $\pm$ 0.03 | -1.42 $\pm$ 0.00 | 2.30 $\pm$ 0.03 | $\mathbf{2.71 \pm 0.02}$ |
| power - $\mathcal{N}(0,1)$ | -0.08 $\pm$ 0.00 | -0.06 $\pm$ 0.00 | $-\mathbf{0.02 \pm 0.00}$ | -0.03 $\pm$ 0.00 |
| NealPrior | -0.05 $\pm$ 0.00 | -0.05 $\pm$ 0.00 | $-\mathbf{0.01 \pm 0.00}$ | $-\mathbf{0.01 \pm 0.00}$ |
| ScalePrior | -0.13 $\pm$ 0.00 | -0.05 $\pm$ 0.00 | $-\mathbf{0.01 \pm 0.00}$ | $-\mathbf{0.01 \pm 0.00}$ |
| protein - $\mathcal{N}(0,1)$ | -1.09 $\pm$ 0.00 | -1.14 $\pm$ 0.00 | $-\mathbf{1.06 \pm 0.01}$ | -1.09 $\pm$ 0.00 |
| NealPrior | -1.11 $\pm$ 0.00 | -1.13 $\pm$ 0.00 | $-\mathbf{1.09 \pm 0.00}$ | $-\mathbf{1.09 \pm 0.00}$ |
| ScalePrior | -1.13 $\pm$ 0.00 | -1.12 $\pm$ 0.00 | $-\mathbf{1.07 \pm 0.00}$ | $-\mathbf{1.07 \pm 0.00}$ |
| wine - $\mathcal{N}(0,1)$ | -1.48 $\pm$ 0.00 | -1.47 $\pm$ 0.00 | $-\mathbf{1.36 \pm 0.00}$ | $-\mathbf{1.36 \pm 0.00}$ |
| NealPrior | -1.31 $\pm$ 0.00 | -1.30 $\pm$ 0.00 | $-\mathbf{1.22 \pm 0.00}$ | -1.23 $\pm$ 0.00 |
| ScalePrior | -1.46 $\pm$ 0.00 | -1.29 $\pm$ 0.00 | $-\mathbf{1.22 \pm 0.00}$ | -1.23 $\pm$ 0.00 |
| yacht - $\mathcal{N}(0,1)$ | -1.04 $\pm$ 0.02 | -1.30 $\pm$ 0.02 | $\mathbf{0.08 \pm 0.01}$ | -0.23 $\pm$ 0.01 |
| NealPrior | -0.46 $\pm$ 0.02 | -0.39 $\pm$ 0.01 | $\mathbf{0.74 \pm 0.01}$ | 0.31 $\pm$ 0.01 |
| ScalePrior | -1.61 $\pm$ 0.00 | -0.77 $\pm$ 0.10 | $\mathbf{0.79 \pm 0.01}$ | 0.30 $\pm$ 0.01 |

Table 7: Average test log likelihoods for our rerun of Salimbeni & Deisenroth (2017), and our implementations of local and global inducing points for deep GPs of various depths.

| {dataset} - {depth} | DSVI | local inducing | global inducing |
|---|---|---|---|
| boston - 2 | -2.50 ± 0.05 | -2.48 ± 0.05 | $-\mathbf{2.35 \pm 0.05}$ |
| 3 | -2.50 ± 0.05 | -2.49 ± 0.05 | $-\mathbf{2.36 \pm 0.05}$ |
| 4 | -2.51 ± 0.05 | -2.49 ± 0.05 | $-\mathbf{2.35 \pm 0.05}$ |
| 5 | -2.52 ± 0.05 | -2.47 ± 0.05 | $-\mathbf{2.35 \pm 0.05}$ |
| concrete - 2 | -3.14 ± 0.01 | -3.13 ± 0.02 | $-\mathbf{3.11 \pm 0.02}$ |
| 3 | -3.15 ± 0.01 | -3.13 ± 0.02 | $-\mathbf{3.11 \pm 0.02}$ |
| 4 | -3.15 ± 0.01 | -3.13 ± 0.02 | $-\mathbf{3.11 \pm 0.02}$ |
| 5 | $-\mathbf{3.11 \pm 0.01}$ | -3.13 ± 0.02 | $-\mathbf{3.11 \pm 0.02}$ |
| energy - 2 | -0.72 ± 0.02 | -0.71 ± 0.03 | $-\mathbf{0.69 \pm 0.03}$ |
| 3 | -0.78 ± 0.02 | -0.71 ± 0.03 | $-\mathbf{0.70 \pm 0.03}$ |
| 4 | -0.88 ± 0.02 | -0.71 ± 0.03 | $-\mathbf{0.70 \pm 0.03}$ |
| 5 | -0.79 ± 0.02 | $-\mathbf{0.70 \pm 0.03}$ | $-\mathbf{0.70 \pm 0.03}$ |
| kin8nm - 2 | $\mathbf{1.34 \pm 0.00}$ | 1.36 ± 0.00 | $\mathbf{1.34 \pm 0.00}$ |
| 3 | $\mathbf{1.38 \pm 0.01}$ | 1.35 ± 0.00 | 1.34 ± 0.00 |
| 4 | $\mathbf{1.38 \pm 0.00}$ | 1.35 ± 0.00 | 1.34 ± 0.00 |
| 5 | $\mathbf{1.36 \pm 0.00}$ | 1.35 ± 0.00 | 1.34 ± 0.00 |
| naval - 2 | 6.55 ± 0.22 | 7.88 ± 0.03 | $\mathbf{7.99 \pm 0.06}$ |
| 3 | 6.46 ± 0.23 | 7.82 ± 0.04 | $\mathbf{8.02 \pm 0.05}$ |
| 4 | 6.79 ± 0.05 | 7.83 ± 0.05 | $\mathbf{8.00 \pm 0.03}$ |
| 5 | 6.70 ± 0.07 | 7.82 ± 0.04 | $\mathbf{7.94 \pm 0.04}$ |
| power - 2 | -2.78 ± 0.01 | $-\mathbf{2.74 \pm 0.01}$ | -2.78 ± 0.01 |
| 3 | -2.74 ± 0.01 | $-\mathbf{2.72 \pm 0.01}$ | -2.78 ± 0.01 |
| 4 | -2.74 ± 0.01 | $-\mathbf{2.73 \pm 0.01}$ | -2.78 ± 0.01 |
| 5 | -2.74 ± 0.01 | $-\mathbf{2.72 \pm 0.01}$ | -2.78 ± 0.01 |
| protein - 2 | $-\mathbf{2.80 \pm 0.00}$ | -2.81 ± 0.00 | -2.82 ± 0.00 |
| 3 | $-\mathbf{2.73 \pm 0.00}$ | -2.78 ± 0.02 | -2.78 ± 0.00 |
| 4 | $-\mathbf{2.71 \pm 0.00}$ | -2.79 ± 0.01 | -2.78 ± 0.00 |
| 5 | $-\mathbf{2.70 \pm 0.00}$ | -2.82 ± 0.00 | -2.78 ± 0.00 |
| wine - 2 | $-\mathbf{0.95 \pm 0.01}$ | -0.96 ± 0.01 | -0.96 ± 0.01 |
| 3 | $-\mathbf{0.95 \pm 0.01}$ | -0.96 ± 0.01 | -0.96 ± 0.01 |
| 4 | $-\mathbf{0.95 \pm 0.01}$ | -0.96 ± 0.01 | -0.96 ± 0.01 |
| 5 | $-\mathbf{0.95 \pm 0.01}$ | -0.96 ± 0.01 | -0.96 ± 0.01 |
| yacht - 2 | -0.41 ± 0.03 | $-\mathbf{0.27 \pm 0.06}$ | -0.29 ± 0.05 |
| 3 | -0.46 ± 0.03 | $-\mathbf{0.23 \pm 0.04}$ | -0.25 ± 0.06 |
| 4 | -0.49 ± 0.03 | $-\mathbf{0.20 \pm 0.05}$ | -0.25 ± 0.05 |
| 5 | -0.49 ± 0.03 | -0.27 ± 0.05 | $-\mathbf{0.23 \pm 0.04}$ |

Table 8: Test RMSEs for our rerun of Salimbeni & Deisenroth (2017), and our implementations of local and global inducing points for deep GPs of various depths.

| {dataset} - {depth} | DSVI | local inducing | global inducing |
|---|---|---|---|
| boston - 2 | $2.94 \pm 0.17$ | $2.87 \pm 0.14$ | $\mathbf{2.63 \pm 0.14}$ |
| 3 | $2.94 \pm 0.18$ | $2.89 \pm 0.14$ | $\mathbf{2.68 \pm 0.14}$ |
| 4 | $2.98 \pm 0.18$ | $2.88 \pm 0.15$ | $\mathbf{2.64 \pm 0.13}$ |
| 5 | $2.99 \pm 0.19$ | $2.87 \pm 0.13$ | $\mathbf{2.63 \pm 0.13}$ |
| concrete - 2 | $5.73 \pm 0.09$ | $5.53 \pm 0.09$ | $\mathbf{5.42 \pm 0.10}$ |
| 3 | $5.75 \pm 0.09$ | $5.54 \pm 0.11$ | $\mathbf{5.46 \pm 0.12}$ |
| 4 | $5.74 \pm 0.09$ | $5.53 \pm 0.11$ | $\mathbf{5.42 \pm 0.10}$ |
| 5 | $5.49 \pm 0.09$ | $5.52 \pm 0.12$ | $\mathbf{5.42 \pm 0.10}$ |
| energy - 2 | $0.49 \pm 0.01$ | $\mathbf{0.48 \pm 0.01}$ | $\mathbf{0.48 \pm 0.01}$ |
| 3 | $0.51 \pm 0.01$ | $\mathbf{0.48 \pm 0.01}$ | $\mathbf{0.48 \pm 0.01}$ |
| 4 | $0.56 \pm 0.02$ | $0.49 \pm 0.01$ | $\mathbf{0.48 \pm 0.01}$ |
| 5 | $0.51 \pm 0.01$ | $\mathbf{0.48 \pm 0.01}$ | $\mathbf{0.48 \pm 0.01}$ |
| kin8nm - 2 | $\mathbf{0.06 \pm 0.00}$ | $\mathbf{0.06 \pm 0.00}$ | $\mathbf{0.06 \pm 0.00}$ |
| 3 | $\mathbf{0.06 \pm 0.00}$ | $\mathbf{0.06 \pm 0.00}$ | $\mathbf{0.06 \pm 0.00}$ |
| 4 | $\mathbf{0.06 \pm 0.00}$ | $\mathbf{0.06 \pm 0.00}$ | $\mathbf{0.06 \pm 0.00}$ |
| 5 | $\mathbf{0.06 \pm 0.00}$ | $\mathbf{0.06 \pm 0.00}$ | $\mathbf{0.06 \pm 0.00}$ |
| naval - 2 | $\mathbf{0.00 \pm 0.00}$ | $\mathbf{0.00 \pm 0.00}$ | $\mathbf{0.00 \pm 0.00}$ |
| 3 | $\mathbf{0.00 \pm 0.00}$ | $\mathbf{0.00 \pm 0.00}$ | $\mathbf{0.00 \pm 0.00}$ |
| 4 | $\mathbf{0.00 \pm 0.00}$ | $\mathbf{0.00 \pm 0.00}$ | $\mathbf{0.00 \pm 0.00}$ |
| 5 | $\mathbf{0.00 \pm 0.00}$ | $\mathbf{0.00 \pm 0.00}$ | $\mathbf{0.00 \pm 0.00}$ |
| power - 2 | $3.87 \pm 0.04$ | $\mathbf{3.74 \pm 0.04}$ | $3.87 \pm 0.04$ |
| 3 | $3.75 \pm 0.04$ | $\mathbf{3.67 \pm 0.04}$ | $3.89 \pm 0.04$ |
| 4 | $3.75 \pm 0.04$ | $\mathbf{3.68 \pm 0.04}$ | $3.87 \pm 0.03$ |
| 5 | $3.73 \pm 0.04$ | $\mathbf{3.67 \pm 0.05}$ | $3.87 \pm 0.04$ |
| protein - 2 | $\mathbf{3.99 \pm 0.02}$ | $4.04 \pm 0.02$ | $4.08 \pm 0.01$ |
| 3 | $\mathbf{3.77 \pm 0.01}$ | $3.96 \pm 0.05$ | $3.95 \pm 0.01$ |
| 4 | $\mathbf{3.70 \pm 0.01}$ | $3.99 \pm 0.05$ | $3.93 \pm 0.01$ |
| 5 | $\mathbf{3.69 \pm 0.01}$ | $4.08 \pm 0.02$ | $3.93 \pm 0.02$ |
| wine - 2 | $\mathbf{0.63 \pm 0.01}$ | $\mathbf{0.63 \pm 0.01}$ | $\mathbf{0.63 \pm 0.01}$ |
| 3 | $\mathbf{0.63 \pm 0.01}$ | $\mathbf{0.63 \pm 0.01}$ | $\mathbf{0.63 \pm 0.01}$ |
| 4 | $\mathbf{0.63 \pm 0.01}$ | $\mathbf{0.63 \pm 0.01}$ | $\mathbf{0.63 \pm 0.01}$ |
| 5 | $\mathbf{0.63 \pm 0.01}$ | $\mathbf{0.63 \pm 0.01}$ | $\mathbf{0.63 \pm 0.01}$ |
| yacht - 2 | $0.42 \pm 0.03$ | $\mathbf{0.34 \pm 0.03}$ | $0.36 \pm 0.03$ |
| 3 | $0.44 \pm 0.03$ | $\mathbf{0.33 \pm 0.03}$ | $0.35 \pm 0.03$ |
| 4 | $0.44 \pm 0.04$ | $\mathbf{0.33 \pm 0.03}$ | $0.35 \pm 0.03$ |
| 5 | $0.45 \pm 0.03$ | $0.35 \pm 0.03$ | $\mathbf{0.33 \pm 0.02}$ |

Table 9: ELBOs per datapoint for our rerun of Salimbeni & Deisenroth (2017), and our implementations of local and global inducing points for deep GPs of various depths.

| {dataset} - {depth} | DSVI | local inducing | global inducing |
|---|---|---|---|
| boston - 2 | -0.48 ± 0.01 | -0.37 ± 0.01 | **−0.29 ± 0.01** |
| 3 | -0.49 ± 0.01 | -0.38 ± 0.01 | **−0.28 ± 0.01** |
| 4 | -0.50 ± 0.01 | -0.37 ± 0.01 | **−0.30 ± 0.01** |
| 5 | -0.52 ± 0.01 | -0.37 ± 0.01 | **−0.29 ± 0.01** |
| concrete - 2 | -0.52 ± 0.00 | -0.42 ± 0.00 | **−0.39 ± 0.00** |
| 3 | -0.53 ± 0.00 | -0.42 ± 0.00 | **−0.40 ± 0.01** |
| 4 | -0.58 ± 0.01 | -0.42 ± 0.00 | **−0.39 ± 0.00** |
| 5 | -0.62 ± 0.00 | -0.43 ± 0.01 | **−0.38 ± 0.00** |
| energy - 2 | 0.97 ± 0.00 | 1.40 ± 0.02 | **1.47 ± 0.00** |
| 3 | 0.86 ± 0.01 | 1.42 ± 0.02 | **1.48 ± 0.00** |
| 4 | 0.73 ± 0.02 | 1.40 ± 0.02 | **1.48 ± 0.00** |
| 5 | 0.82 ± 0.02 | 1.44 ± 0.01 | **1.48 ± 0.00** |
| kin8nm - 2 | -0.18 ± 0.00 | **−0.16 ± 0.00** | **−0.16 ± 0.00** |
| 3 | -0.18 ± 0.00 | **−0.16 ± 0.00** | **−0.16 ± 0.00** |
| 4 | -0.18 ± 0.00 | **−0.16 ± 0.00** | **−0.16 ± 0.00** |
| 5 | -0.19 ± 0.00 | **−0.16 ± 0.00** | **−0.16 ± 0.00** |
| naval - 2 | 2.02 ± 0.27 | 3.34 ± 0.04 | **3.63 ± 0.04** |
| 3 | 1.87 ± 0.31 | 3.29 ± 0.05 | **3.57 ± 0.06** |
| 4 | 2.30 ± 0.06 | 3.25 ± 0.06 | **3.54 ± 0.06** |
| 5 | 2.17 ± 0.08 | 3.32 ± 0.04 | **3.51 ± 0.05** |
| power - 2 | 0.02 ± 0.00 | **0.05 ± 0.00** | 0.03 ± 0.00 |
| 3 | 0.03 ± 0.00 | **0.05 ± 0.00** | 0.03 ± 0.00 |
| 4 | 0.03 ± 0.00 | **0.05 ± 0.00** | 0.04 ± 0.00 |
| 5 | 0.03 ± 0.00 | **0.05 ± 0.00** | 0.03 ± 0.00 |
| protein - 2 | **−1.06 ± 0.00** | -1.07 ± 0.00 | **−1.06 ± 0.00** |
| 3 | **−1.02 ± 0.00** | -1.05 ± 0.01 | -1.05 ± 0.00 |
| 4 | **−1.01 ± 0.00** | -1.06 ± 0.01 | -1.04 ± 0.00 |
| 5 | **−1.01 ± 0.00** | -1.08 ± 0.00 | -1.04 ± 0.00 |
| wine - 2 | -1.19 ± 0.00 | -1.18 ± 0.00 | **−1.17 ± 0.00** |
| 3 | -1.19 ± 0.00 | -1.18 ± 0.00 | **−1.17 ± 0.00** |
| 4 | -1.20 ± 0.00 | -1.18 ± 0.00 | **−1.17 ± 0.00** |
| 5 | -1.20 ± 0.00 | -1.18 ± 0.00 | **−1.17 ± 0.00** |
| yacht - 2 | 1.16 ± 0.01 | 1.78 ± 0.01 | **1.86 ± 0.01** |
| 3 | 1.09 ± 0.01 | 1.80 ± 0.01 | **1.91 ± 0.01** |
| 4 | 1.07 ± 0.01 | 1.81 ± 0.01 | **1.91 ± 0.01** |
| 5 | 1.07 ± 0.01 | 1.81 ± 0.01 | **1.89 ± 0.01** |

