# OpenReview forum: "Global inducing point variational posteriors for Bayesian neural networks and deep Gaussian processes"
_ICLR.cc/2021/Conference — Reject_

### Official Review · AnonReviewer2 · 2020-10-25
**Clever idea but not well explained about how it actually works**

**Rating:** 7
**Confidence:** 3

**Review:**

The paper proposed a new way of doing Bayesian deep learning in which the optimal conditional posterior for the last layer weights could be reached if the inducing input $Z_0$ is chosen to be the input data $X$ and the pseudo-observation for the last layer $V^L$ is the observation $Y$. Instead of factorizing the inducing points The global inducing input $Z_0$ is propagated through the network to ensure posterior dependencies across layers. The authors also extend this idea to deep Gaussian processes so that the latent functions across layers are correlated. Experiments show better performance than previous methods on both synthetic and real datasets, without the need to anneal the weights for the KL term in ELBO.

Pros:
The idea of capturing the posterior dependencies between layers in BNN and DGP is great, and may have a good amount of significance in both areas.

Cons:
The description of the method itself is not very clear. For the BNN part, while Alg. 1 is easy to follow, how does it relates to the training process of BNN? The DGP part is even harder to understand -- what is the complete form of the variational distribution and the ELBO? How is the ELBO optimized? The paper didn't say much about these important details, and by just staring at Eq. (16) I really couldn't figure them out. Comparing to methods the paper compares with such as (Salimbeni and Deisenroth, 2017), Section 3 of this paper is not clearly written.

Other questions:

How does $N_{l-1}$ in Eq. (2) disappear in Eq. (14)?

Does this method apply to GP classification?

In Sec. 4.1, 100 inducing points per layer was used on a dataset with only 40 points. What if the number of inducing points is smaller? Does the proposed method still work as expected?


-----------------------

I thank the authors for their detailed reply. The revision after the initial review solves the concerns and questions I had. Very good work. I vote for accept.

---

> ### Author Response · Authors · 2020-11-16
> **Response to review**
>
> Thank you for your feedback!
>
> Once the ELBO is computed using Alg. 1, we optimise the ELBO using reparameterised variational inference (see new Appendix A).
>
> We have added Eqs.10 with the full form for the ELBO for BNNs.  For DGPs, we have considerably expanded Appendix C (now Appendix D) with the full form for the approximate posterior (Eq. 60) and ELBO (Eq. 63). We have also added an Algorithm, similar to Alg. 1, for DGPs in the Appendix. Finally, we have modified Section 3 so that it is hopefully clearer, including explicitly stating the variational posterior and ELBO.
>
> Responses to your other questions:
> $N_{l-1}$ in Eq. 14 (now Eq. 17) should not have disappeared. Thank you for pointing this out! It is now fixed.
>
> Our method can be applied with any likelihood function and so can in fact be used for (D)GP classification: we did DGP regression in Appendix J (Fig. A4), and we did BNN classification in the main text on CIFAR-10.  The classification top layer and the DGP lower layers can readily be combined.
>
> We have added appendix H, which includes a figure (Fig. A2) that shows the effect of having different numbers of inducing points, ranging from 2 to 40. Our method still performs remarkably well for just 10 points.
>
> If parts of the text can still be clarified we would be happy to address these issues.

---

### Official Review · AnonReviewer1 · 2020-11-04
**Approximate posterior taking into account correlations across the layers**

**Rating:** 7
**Confidence:** 3

**Review:**

**Summary**:
The paper proposes a posterior estimation for Bayesian neural networks (BNNs) and deep Gaussian processes (DGPs). The difference from the previous approaches is to use global inducing points which help to take into account correlations across layers.

I find a paper well written and the results worth publishing.

**Comments**:
“We found that the ELBO for methods that factorise across layers — factorised and local inducing — drops rapidly as networks get deeper and wider (Fig. 2). This is undesirable behaviour, as we know that wide, deep networks are necessary for good performance on difficult machine learning tasks. In contrast, we found that methods with global inducing points at the last layer decay much more slowly with depth, and perform better as networks get wider.” I wonder if it can be due to the increasing tails with depth in BNNs with Gaussian prior as stated by Vladimirova et al. (2019). According to their paper, the distributions on the units across the layers become heavier-tailed but knowing that the limit is Gaussian Process (Matthews et al., 2018; Lee et al., 2018), the distributions on the units for wider neural networks are closer to Gaussians (I think the center of the distribution reminds the Gaussian more and more). And as the methods that factorise across the layers use the assumptions of Gaussian units, the error is higher for shallow NNs. So maybe it is possible to keep in mind the “heaviness” for shallow NNs induces due to the correlations between the units and to improve the results.

Lee et al. (2018) Deep neural networks as Gaussian processes, ICLR

Matthews et al. (2018) Gaussian Process Behaviour in Wide Deep Neural Networks, ICLR

Vladimirova et al. (2019) Understanding priors in Bayesian neural networks at the unit level, ICML



**Minor**:
- “initalization”
- I think you should stick either to American, either to British version of writing: “factorises”, “parameterise”, “initialization”, “initialize”
- “work using using infinite-width neural networks”
- “The next block show”
- “it is difficult design” -> to
- “downweighted”
- “by coupling inducing inputs to inducing outputs” -> to induce
- “the number of locations within an patch”
- “we need to be able sample”
- “these issues do not arises”

---

> ### Author Response · Authors · 2020-11-16
> **Response to review**
>
> Thank you for your review and feedback!
>
> Interestingly, the infinite width neural network results of Lee et al. (2018) and Matthews et al. (2018) were a huge inspiration for this work.  In particular, the posterior over the lower layers in these models is actually equal to the prior (Hron et al. 2020 "Exact posterior distributions of wide Bayesian neural networks"),  with the top-layer posterior being in effect Bayesian linear regression.  This was interesting from the perspective of variational inference, because if the approximate posterior equals the prior for the lower layers, then the KL-divergence to the prior for those layers is 0 and as the ELBO can be written as $ELBO = E[log P(y|x, w) - D_{KL}(Q(W)|| P(W))]$, we can have arbitrarily deep networks without hurting the ELBO.  This doesn't happen in standard factorised networks: the approximate posterior and prior are different at all layers, implying that (all else being equal) the ELBO should get worse as the network gets deeper.  To obtain an approximate posterior over the top layer which gave good performance despite the lower layers being drawn at random from the prior, we found that we needed to use the optimal top-layer posterior, which corresponded to the global inducing posterior.
>
> An investigation of the degree and performance implications of heavy-tailed behaviour in the activities of variational Bayesian neural networks with different approximate posteriors would be exciting follow-up work.  We speculate that:
> * following Vladimirova et al. (2019), the activities of neurons in a factorised network become more heavy-tailed with depth, and
> * the design of our approximate posterior might actually suppress heavy-tailed activations.  In particular, at each layer, we sample the weights by in effect conditioning on the outputs being close to some target.  That conditioning uses a Gaussian likelihood which may well encourage the activations to be more Gaussian and less heavy-tailed.
> It would be interesting to investigate whether these differences in heavy-tailed behaviour might be one way to understand the performance differences between global inducing and factorised methods.
>
> Thank you for pointing out the spelling/grammatical errors! These should now be fixed.

---

### Official Review · AnonReviewer6 · 2020-11-06
**see review**

**Rating:** 6
**Confidence:** 3

**Review:**

This paper proposes a posterior approximation for BNN that models correlations between the layers weights. The paper begins by pointing out that, for any posterior distribution approximation, the optimal conditional posterior distribution over the top-layer weights given the weights of the previous layers has a closed-form in the case of Gaussian likelihood, which, due to Bayes rule, turns out to be product of the likelihood and the top-layer prior. Based on this insight the paper proposes to model each conditional posterior distribution over intermediate layers weights given previous layers weights following the same structure, that is, as a product of a 'pseudo-likelihood' over unobserved noisy activations and the prior for that layer. In order to make inference tractable, the paper proposes the use of global inducing points as well as noisy pseudo-observations of the activations of intermediate layers which are treated as variational parameters.  The paper also describes how such procedure applies to convolutional neural networks and how it can be applied for DGPs as well.

Even though the paper covers several topics, its presentation is clear and straightforward. The main issue I see with the paper is that the validation of the proposed model is done mostly in terms of accuracy and log-likelihood. As pointed out by [1], one of the main interests on posterior approximations that allow correlations between the weights is for capturing the uncertainty of the compositional structure. Based on this, I think the paper could increase its strength by describing the advantages on interpretability of the learned approximate posterior provides over a fully independent posterior approximation such as [2]. In particular, it would be useful to see how the conditional posterior distributions evolve given the previous layers weights.

[1] Ivan Ustyuzhaninov, Ieva Kazlauskaite, Markus Kaiser, Erik Bodin, Neill DF Campbell, and Carl Hen-
rik Ek. Compositional uncertainty in deep Gaussian processes. UAI, 2020.

[2] Hugh Salimbeni and Marc Deisenroth. Doubly stochastic variational inference for deep Gaussian
processes. In Advances in Neural Information Processing Systems, pp. 4588–4599, 2017.

---

> ### Author Response · Authors · 2020-11-16
> **Response to review**
>
> We have added Appendix I, inspired by Ustyuzhaninov 2020, outlining how global inducing points allow us to capture uncertainty across the compositional structure. In particular, we show empirically that global inducing does not suffer from the collapse onto deterministic functions that Ustyuzhaninov et al demonstrate happens for methods that are factorised across layers, as in local inducing. We have attempted to show this for both DGPs and BNNs. Moreover, our previous empirical results speak to our success in accurately modelling posterior across layers.  In particular, we can write the ELBO as $log P(y| x) - D_{KL}(Q(W)|| P(W|y,x))$.  Thus, our empirical improvements to the ELBO (Table 1 and Fig 3, top) imply improvements in accurately modelling posteriors over weights across layers.
>
> Your suggestion about interpretability is very exciting, and we're interested in writing follow-up work on it! There is a possibility that our approximate posterior is indeed more interpretable, in particular because it is phrased in terms of "target" outputs or feature maps at every layer for each inducing input image.  As such, it should be possible to draw the target feature maps and compare with the corresponding input image.  That said, interpretability is a highly nontrivial subject, so we feel that it is only possible to do it justice as a standalone paper.
>
> If you have any suggestions for experiments that we could run in addition to those in Appendix I, we would love to hear them!

---

### Author Response · Authors · 2021-01-20
**Response**

The reviewers comprehensively assessed our paper, and universally recommended acceptance with scores of 7/7/6.  This placed our paper in the top 13% of submitted papers, compared to a 29% acceptance rate.  Indeed 3 papers with the exact same scores (7/7/6) were awarded Spotlights.

As such, we were surprised and confused to find the AC had ignored the considerable efforts of the reviewers, and opted to reject our paper.  The stated reasoning was primarily a supposed lack of a comparison to SGHMC.  But had the AC even glanced at our paper, they would have seen that this comparison was very prominently included in Table 1 of the main text of the original submission (Wenzel et al. 2020).

---

### Decision · Program_Chairs · 2021-01-07
**Final Decision**

**Decision:**

Reject

**Comment:**

The paper proposes a variational inference method for Bayesian neural networks where the approximate posterior models the correlations between the weights at all layers, using the concept of “global” inducing points.

Some concerns raised by the reviewers regarding how global inducing points allow us to capture uncertainty across the compositional structure and clarity of the manuscript have been addressed by the authors. However, the more general issue of interpretability has been left for future work.

One of the main deficiencies of this paper is that it seems to ignore other scalable approaches that also provide more complex posteriors, for example, those based on stochastic gradient Hamiltonian Monte Carlo (see, e.g., https://arxiv.org/abs/1806.05490, and references therein). Overall, although there was support for this paper, it is unclear if approaches such as those presented here are really necessary. A comparison between the two methodologies maybe not only illustrative but required.